# CAUSALLM IS NOT OPTIMAL FOR IN-CONTEXT LEARNING

**Nan Ding**    **Tomer Levinboim**    **Jialin Wu**    **Sebastian Goodman**    **Radu Soricut**
Google Research
{dingnan,tomerl,jialinwu,seabass,rsoricut}@google.com

## ABSTRACT

Recent empirical evidence indicates that transformer based in-context learning performs better when using a prefix language model (prefixLM), in which in-context samples can all attend to each other, compared to causal language models (causalLM), which use auto-regressive attention that prohibits in-context samples to attend to future samples. While this result is intuitive, it is not understood from a theoretical perspective. In this paper we take a theoretical approach and analyze the convergence behavior of prefixLM and causalLM under a certain parameter construction. Our analysis shows that both LM types converge to their stationary points at a linear rate, but that while prefixLM converges to the optimal solution of linear regression, causalLM convergence dynamics follows that of an online gradient descent algorithm, which is not guaranteed to be optimal even as the number of samples grows infinitely. We supplement our theoretical claims with empirical experiments over synthetic and real tasks and using various types of transformers. Our experiments verify that causalLM consistently underperforms prefixLM in all settings.

## 1 INTRODUCTION

Transformer-based models (Vaswani et al., 2017) have become the default foundational model for various machine learning applications such as natural language processing (Devlin et al., 2018; Brown et al., 2020; Chowdhery et al., 2022) and computer vision (Dosovitskiy et al., 2020). Beyond their traditional usage in machine learning applications, it has recently been discovered that pretraining large transformers on a vast amounts of data leads them to develop a striking ability referred to as in-context learning (ICL) (Brown et al., 2020). Specifically, once such pretraining is complete, these models are able to solve new tasks at inference time (without changing their parameters) by simply ingesting a short sequence (prefix) of labeled examples from a task and then computing a prediction for a query example.

The ICL capability was first demonstrated by GPT-3 (Brown et al., 2020), where a causalLM (a Transformer decoder with auto-regressive attention masks) was used as the main model architecture. However, follow up work empirically found that restricting the auto-regressive masks on the entire sequence is too prohibitive and therefore proposed the so-called prefixLM (Raffel et al., 2020b; Tay et al., 2022) which allows full attention within the prefix tokens. Moreover, the latest models (such as PaLM2 (Google et al., 2023)) adopt a mixture of different LM objectives during pretraining to achieved state-of-art performance across a diverse set of tasks and capabilities.

However, beyond the few empirical results in those and related papers, there is yet no theoretical explanation that accounts for the different ICL behavior of prefixLM and causalLM. Indeed, theoretical studies of ICL are difficult due to the complicated non-linearity of the (ordinary) transformer architecture. However, recent work (Von Oswald et al., 2023) focusing on ICL of linear regression was able to show that a specifically designed parameter construction of a one-layer Linear Self-Attention (LSA) transformer can simulate a single step of gradient descent by using the in-context examples as training data. Moreover, a different recent study (Zhang et al., 2023) used gradient flow to prove that a randomly initialized LSA-transformer indeed converges to such a construction during training.

In this paper, we continue the theoretical line of work above by investigating the convergence properties of ICL for both prefixLM and causalLM multi-layer LSA-transformers in a linear regression setting. We summarizes our contributions as follows:

- We first present a clear, formal proof that establishes the relationship between a multi-layer LSA and multi-step gradient descents in linear regression.

- We then show that both causalLM and prefixLM based multi-layer LSA-transformers converge to their respective stationary points with linear rates of convergence. We prove that the stationary point of prefixLM corresponds to the optimal least square solution of the linear regression problem, while the stationary points of causalLM correspond to the weights obtained along the iterations of online gradient descent with non-decaying step sizes. Importantly, the stationary points obtained by causalLM may not become optimal even as the number of in-context examples increases, which indicates that causalLM is not optimal for in-context learning.

- Finally, we verify the above theoretical insights by conducting experiments with LSA-transformers as well as ordinary softmax attention based transformers on various synthetic tasks including linear and non-linear regression, and multiclass classifications. We also compare causalLM and prefixLM ICL based on LLMs including T5 (Roberts et al., 2022) and PaLM2 (Google et al., 2023), as well as the multimodal model PaLI-X (Chen et al., 2023). Our experimental results support our theoretical findings and consistently show the superiority of prefixLM over causalLM on ICL for such settings.

## 2 BACKGROUND

We begin by reviewing a few types of transformer attention and in-context learning (ICL), as well as a specific transformer construction for linear regression ICL by (Von Oswald et al., 2023) which our theories will be based on. The discussions of other related work are deferred to Appendix A.

### 2.1 TRANSFORMERS: SSA, LSA, CAUSALLM, AND PREFIXLM

Given a sequence of input vectors $\mathbf{Z} = (\mathbf{z}_1, \ldots, \mathbf{z}_n)$, the output of standard Softmax Self-Attention (SSA) layer is

$$\mathbf{z}_j \leftarrow \mathbf{z}_j + \mathbf{P}\,\mathbf{V}\,\mathbf{Z}\,\mathrm{softmax}(\mathbf{Z}^\top\,\mathbf{K}^\top\,\mathbf{Q}\,\mathbf{z}_j),$$

where $\mathbf{P}, \mathbf{V}, \mathbf{K}, \mathbf{Q}$ respectively corresponds to the output projection, value transformation, key transformation and query transformation.

Since the softmax attention of standard transformers is non-linear, its theoretical analysis becomes complicated even for a single layer. For this reason, theoretical approaches to analyze transformers have often resorted to the Linear Self-Attention (LSA) layer (Von Oswald et al., 2023; Zhang et al., 2023), which simply drops the softmax function from the attention,

$$\mathbf{z}_j \leftarrow \mathbf{z}_j + \mathbf{P}\,\mathbf{V}\,\mathbf{Z}(\mathbf{Z}^\top\,\mathbf{K}^\top\,\mathbf{Q}\,\mathbf{z}_j) = \mathbf{z}_j + \mathbf{P}\,\mathbf{V}\sum_{i=1}^{n}\mathbf{z}_i\left(\mathbf{z}_i^\top\,\mathbf{K}^\top\,\mathbf{Q}\,\mathbf{z}_j\right). \tag{1}$$

Furthermore, since each input $\mathbf{z}_j$ can attend to all positions $j \in \{1 \ldots n\}$, this form of attention is categorized as full (or bidirectional) attention, and is typically used in the transformer encoder.

On the other hand, a (linear) transformer decoder uses the *auto-regressive* attention

$$\mathbf{z}_j \leftarrow \mathbf{z}_j + \mathbf{P}\,\mathbf{V}\sum_{i=1}^{j}\mathbf{z}_i\left(\mathbf{z}_i^\top\,\mathbf{K}^\top\,\mathbf{Q}\,\mathbf{z}_j\right). \tag{2}$$

which restricts each token $\mathbf{z}_j$ to attend only to previous positions (and itself) from $\{1 \ldots j\}$. This restriction is due to the role of the decoder as a causal language model (causalLM) which predicts the next token in the context of the previously generated ones.

The original transformer involves both a full attention based encoder and an auto-regressive attention based decoder. However, prominent NLP research has often chosen either encoder-only (e.g.

BERT (Devlin et al., 2018)) or decoder-only (e.g. GPT (Brown et al., 2020), PaLM (Chowdhery et al., 2022)) models according to the task at hand. This is partially for the purpose of halving the parameter sizes.

Another version of attention, between full and auto-regressive, followed from the observation that some tasks can benefit from a prefix sequence such as context or prompt. That is, the input sequence $\mathbf{Z}$ is composed of $n'$ prefix tokens $(\mathbf{z}_1, \ldots, \mathbf{z}_{n'})$ configured for the task, while the tokens $(\mathbf{z}_{n'+1}, \ldots, \mathbf{z}_n)$ represent the sample. Specifically, prefixLM (Raffel et al., 2020b) suggests the following attention (in its LSA version):

$$\mathbf{z}_j \leftarrow \mathbf{z}_j + \mathbf{P} \, \mathbf{V} \sum_{i=1}^{\max(j,n')} \mathbf{z}_i \left( \mathbf{z}_i^\top \, \mathbf{K}^\top \, \mathbf{Q} \, \mathbf{z}_j \right),$$

where $\max(j, n')$ ensures each prefix token $\mathbf{z}_j$ with $j < n'$ can attend to all prefix tokens.

## 2.2 IN-CONTEXT LEARNING

A formal framework of in-context learning has been described in various existing literature such as (Garg et al., 2022; Zhang et al., 2023). Here, we briefly review the problem setting and introduce notation that will be used across the paper.

In-context learning refers to the ability of models to produce context-driven predictions at inference time. That is, at inference time, a model is fed with a sequence consisting of input-label pairs and a query input $(\mathbf{x}_1, y_1, \ldots, \mathbf{x}_n, y_n, \mathbf{x}_{query})$ and its goal is to predict the label $y_{query}$ of $\mathbf{x}_{query}$ using the context examples $(\mathbf{x}_1, y_1, \ldots, \mathbf{x}_n, y_n)$ (specifically, without changing the model parameters).

## 2.3 LINEAR REGRESSION IN-CONTEXT LEARNERS

Linear regression is a classical machine learning problem. Given a set of input-label pairs $(\mathbf{x}_i, y_i)$, the goal is to find an optimal weight vector $\mathbf{w}$ that minimizes the l2-loss:

$$L(\mathbf{w}) = \frac{1}{2n} \sum_{i=1}^{n} \| \mathbf{w} \, \mathbf{x}_i - y_i \|_2^2.$$

The gradient of the loss is $\nabla_{\mathbf{w}} L = \frac{1}{n} \sum_{i=1}^{n} (\mathbf{w} \, \mathbf{x}_i - y_i) \, \mathbf{x}_i^\top$, and a gradient descent algorithm with step size $\eta$ follows the update rule:

$$\mathbf{w}^{(l)} = \mathbf{w}^{(l-1)} + \frac{\eta}{n} \sum_{i=1}^{n} (y_i - \mathbf{w}^{(l-1)} \, \mathbf{x}_i) \, \mathbf{x}_i^\top . \tag{3}$$

Using linear regression as a lens to study in-context learning was first proposed in (Garg et al., 2022), where the authors laid out an approach for training transformers to in-context learn a class of simple predictors, including linear regression. However, no theoretical study was provided. More recently, and most relevant to our work, (Von Oswald et al., 2023) proposed a succinct construction that demonstrates how a single LSA layer can effectively implement a single gradient descent step. According to their setup the input is formulated as

$$\mathbf{Z} = (\mathbf{z}_1^{(0)}, \ldots, \mathbf{z}_n^{(0)}), \text{ where } \mathbf{z}_j^{(0)} = \begin{pmatrix} \mathbf{x}_j \\ y_j \end{pmatrix} \tag{4}$$

and the parameter matrices of (1) are set as:

$$\mathbf{K} = \mathbf{Q} = \begin{pmatrix} \mathbf{I}_{d \times d} & \mathbf{0} \\ 0 & 0 \end{pmatrix}, \mathbf{V} = \begin{pmatrix} \mathbf{0}_{d \times d} & \mathbf{0} \\ \mathbf{w}^{(0)} & -1 \end{pmatrix}, \mathbf{P} = \frac{\eta}{n} \mathbf{I}, \tag{5}$$

where $\mathbf{w}^{(0)}$ is an initial weight vector. (Von Oswald et al., 2023) then showed that this configuration results in an update of their so-called transformed target $y_j \leftarrow y_j + \eta \left( \nabla_{\mathbf{w}^{(0)}} L \right) \mathbf{x}_j$, and that this target update is equivalent to the one performed by a single-step gradient descent of linear regression.

Although the construction of (Von Oswald et al., 2023) connected LSA-based ICL to the gradient descent of linear regression, the "transformed target" view seems unnatural* to work with. Moreover, their extension from single-layer to multi-layer LSA is unfortunately unclear.

---

*The traditional ML formulation updates the weight vector or the model prediction, while the groundtruth target remains fixed.

## 3   Multi-layer in-context learner

In this section, we provide a formal proof that a multi-layer LSA under the construction of (Von Oswald et al., 2023) progresses identically to multi-step gradient descent.

Instead of the "transformed target" view, the following proposition explicitly connects the GD weights of (3) to the outputs of the multi-layer LSA under the constructions of $\mathbf{K}$, $\mathbf{Q}$, $\mathbf{P}$ and $\mathbf{V}$ in (5). Note that we keep $\mathbf{w}^{(0)} = 0$ in the proposition because it simplifies the equations and makes the outputs more meaningful. However, such specification is not mandatory, and we provide general propositions, for arbitrary $\mathbf{w}^{(0)}$, in Appendix C.

**Proposition 1** *For a multi-layer LSA satisfying the construction* (5) *and with* $\mathbf{w}^{(0)} = 0$, *if its input* $\mathbf{Z}$ *is formatted as* (4), *then its l-th layer output is* $\mathbf{z}_j^{(l)} = (\mathbf{x}_j^\top, \delta_j^{(l)})^\top$, *where* $\delta_j^{(l)} = y_j - \mathbf{w}^{(l)} \mathbf{x}_j$ *and* $\mathbf{w}^{(l)}$ *is the l-th updated weight from the gradient descents update rule in* (3).

*Proof Sketch:* Plugging in $\mathbf{K}$, $\mathbf{Q}$, $\mathbf{P}$ and $\mathbf{V}$ of (5) with $\mathbf{w}^{(0)} = 0$ and $\mathbf{z}_j^{(l)} = (\mathbf{x}_j^\top, \delta_j^{(l)})^\top$ into (1), we obtain that for all $l > 0$,

$$\begin{pmatrix} \mathbf{x}_j \\ \delta_j^{(l)} \end{pmatrix} = \begin{pmatrix} \mathbf{x}_j \\ \delta_j^{(l-1)} \end{pmatrix} - \frac{\eta}{n} \sum_{i=1}^n \begin{pmatrix} \mathbf{0} \\ \delta_i^{(l-1)} \end{pmatrix} \mathbf{x}_i^\top \, \mathbf{x}_j \, .$$

Since $\mathbf{z}_j$ never changes its first $d$-dimension corresponding to $\mathbf{x}_j$, we can simplify it and focus only on $\delta_j^{(l)}$, which is the last output coordinate of the $j$-th LSA-layer,

$$\delta_j^{(l)} = \delta_j^{(l-1)} - \frac{\eta}{n} \sum_{i=1}^n \delta_i^{(l-1)} \mathbf{x}_i^\top \, \mathbf{x}_j, \tag{6}$$

with $\delta_j^{(0)} = y_j$. Defining $\tilde{y}_j^{(l)} = y_j - \delta_j^{(l)}$ and rearranging (6), we obtain $\tilde{y}_j^{(0)} = 0$ and $\forall l > 0$:

$$\tilde{y}_j^{(l)} = \tilde{y}_j^{(l-1)} + \frac{\eta}{n} \sum_{i=1}^n (y_i - \tilde{y}_i^{(l-1)}) \mathbf{x}_i^\top \, \mathbf{x}_j \, . \tag{7}$$

Finally, using (7) and the fact that $\tilde{y}_j^{(0)} = 0 = \mathbf{w}^{(0)} \mathbf{x}_j$, it can be proved by induction that $\forall l : \tilde{y}_j^{(l)} = \mathbf{w}^{(l)} \mathbf{x}_j$. A complete proof is provided in Appendix B.

To summarize, the newly introduced variable $\tilde{y}_j^{(l)}$ is exactly the prediction of the $l$-th gradient descent weights $\mathbf{w}^{(l)}$ for $\mathbf{x}_j$, and $\delta_j^{(l)}$ is the difference between the true label $y_j$ and the predicted $\tilde{y}_j^{(l)}$. Therefore, $\tilde{y}_j^{(l)}$ serves as a bridge to connect the LSA output $\delta_j^{(l)}$ and the GD weight $\mathbf{w}^{(l)}$.

So far, we have dealt with the behavior of LSA layers with full attention. In what follows, we move on to the practical setting of in-context learning, where the input contains not only $n$ in-context (training) examples in the format of (4), but also an additional (test) query $\mathbf{z}_{query}^{(0)} = (\mathbf{x}_{query}^\top, 0)^\top$. In particular, we will focus on the two common ICL variants: prefixLM and causalLM, each with a different type of attention.

### 3.1   PrefixLM ICL

A prefixLM ICL treats the in-context examples $\mathbf{Z}$ as the prefix and uses full attention on the first $n$ positions, so that they can each freely attend to each other. The last query vector $\mathbf{z}_{query}$ can also attend to any example in $\mathbf{Z}$, but cannot attend to itself[†]. As a result, the updates of the prefixLM-ICL under the same construction follow (6), with the outputs of the $l$-th layer being,

$$\delta_j^{(l)} = y_j - \tilde{y}_j^{(l)} = y_j - \mathbf{w}^{(l)} \mathbf{x}_j,$$

$$\text{and} \ \ \delta_{query}^{(l)} = -\tilde{y}_{query}^{(l)} = -\mathbf{w}^{(l)} \mathbf{x}_{query},$$

---

[†]This is because the query does not contain a meaningful label. Attending to itself would cause it to include its last-dim input as a label, which would contaminate the resulting multi-layer prediction. This observation was not considered in (Von Oswald et al., 2023).

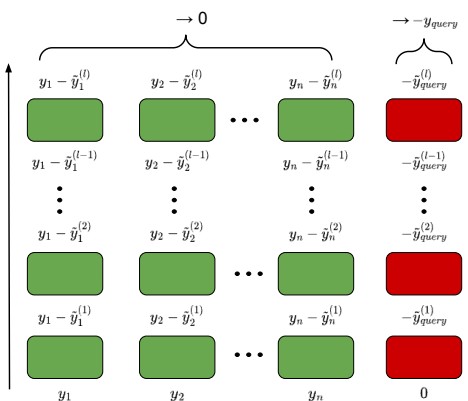

Figure 1: The inputs/outputs of a multi-layer in-context learner. We omitted $\mathbf{x}_j$ and $\mathbf{x}_{query}$ since they are unchanged.

where the initial $\tilde{y}_j^{(0)} = \tilde{y}_{query}^{(0)} = 0$.

Intuitively, the dynamics of the prefixLM ICL is as follows: all $\tilde{y}_j^{(l)}$ starts as 0 at $l = 0$, and gradually approach to the true label $y_j$ as $l$ increases, so that the difference (also as the output) $\delta_j^{(l)}$ gradually approaches to 0. At the same time, $\delta_{query}^{(l)}$ starts at 0, and gradually approaches to $-y_{query}$, the negation of the query label. Figure 1 provides an illustration of these dynamics.

## 3.2 CAUSALLM ICL

A causalLM applies auto-regressive attention throughout the entire sequence. Therefore, plugging the same $\mathbf{K}, \mathbf{Q}, \mathbf{P}, \mathbf{V}$ into (2), the update rules of (6) and (7) become:

$$\delta_j^{(l)} = \delta_j^{(l-1)} - \frac{\eta}{n} \sum_{i=1}^{j} \delta_i^{(l-1)} \mathbf{x}_i^\top \mathbf{x}_j, \tag{8}$$

$$\tilde{y}_j^{(l)} = \tilde{y}_j^{(l-1)} + \frac{\eta}{n} \sum_{i=1}^{j} (y_i - \tilde{y}_i^{(l-1)}) \mathbf{x}_i^\top \mathbf{x}_j \tag{9}$$

‡with $\delta_j^{(l)} = y_j - \tilde{y}_j^{(l)}$. Moreover, since different $\delta_j, \tilde{y}_j$ are exposed to different ranges of inputs, there is no uniform $\mathbf{w}$ as in (3) that is associated with all $\tilde{y}_j$. Instead, if we define $\mathbf{w}_j$ for each different position $j$ with $\mathbf{w}_j^{(0)} = 0$ and

$$\mathbf{w}_j^{(l)} = \mathbf{w}_j^{(l-1)} + \frac{\eta}{n} \sum_{i=1}^{j} (y_i - \mathbf{w}_i^{(l-1)} \mathbf{x}_i) \mathbf{x}_i^\top \tag{10}$$

then we have the following proposition:

**Proposition 2** *For a multi-layer causalLM-LSA satisfying (5) with $\mathbf{w}^{(0)} = 0$, if its input $\mathbf{Z}$ is formatted as (4), then its l-th layer output is $\mathbf{z}_j^{(l)} = (\mathbf{x}_j^\top, \delta_j^{(l)})^\top$, where $\delta_j^{(l)} = y_j - \mathbf{w}_j^{(l)} \mathbf{x}_j$ and $\mathbf{w}_j^{(l)}$ follow (10).*

The proof of Proposition 2 is provided in Appendix B. Similar to prefixLM-ICL, causalLM-ICL also has $\tilde{y}_j^{(l)} = \mathbf{w}_j^{(l)} \mathbf{x}_j$, and

$$\delta_{query}^{(l)} = -\tilde{y}_{query}^{(l)} = -\mathbf{w}_n^{(l)} \mathbf{x}_{query} .$$

In summary, causalLM-ICL and prefixLM-ICL are associated with different update rules: $\mathbf{w}_j^{(l)}$ follows (10) while $\mathbf{w}^{(l)}$ follows (3). Specifically, in causalLM, it can be seen that the $\mathbf{w}_i^{(l-1)}$

---

‡There is another way of update which changes $\eta/n$ to $\eta/j$ for the $j$-th example. We provide more details in Appendix D and show it performs worse than the main version in (8).

corresponding to the first positions are biased due to restricted access to only a few data points and furthermore, that these biases are propagated to later positions by (10). In prefixLM on the other hand, each position has access to all the data and a single $\mathbf{w}^{(l)}$ can be used across the entire sequence as in (3). Although Eq. (3) and Eq. (10) only hold for the structured LSA case, the profound difference between causalLM and prefixLM stems from their architectural difference and therefore we believe extends to general transformers, as indicated by our experimental results in Section 5.

## 4 CONVERGENCE OF THE MULTI-LAYER IN-CONTEXT LEARNERS

In this section, we prove that both multi-layer prefixLM and causalLM converge to their respective stationary points with increasing layers (and with linear rates). In addition, we show that the stationary point of prefixLM corresponds to the optimal least-square solution of the linear regression problem, while the ones corresponding to causalLM are equivalent to the iterative weights of online gradient descent of linear regression, which are known to be sub-optimal for a limited number of examples.

### 4.1 CONVERGENCE OF THE PREFIXLM ICL

The fact that a multi-layer prefixLM computation exactly follows the update rule of $\mathbf{w}^{(l)}$ as in (3), implies that the layer outputs of prefixLM have the same dynamics of multi-step gradient descent on a linear regression problem. The convergence properties of such dynamics are well-known, and are stated in the following proposition:

**Proposition 3** *If $\mathbf{w}^{(l)}$ follows the iterative updates of* (3)*, then there exists a stationary point $\mathbf{w}^*$ with coefficients satisfying:*

$$\mathbf{y}\,\mathbf{X}^\top = \mathbf{w}^*\,\mathbf{X}\,\mathbf{X}^\top,$$

*where $\mathbf{y} = (y_1, \ldots, y_n)$ and $\mathbf{X} = (\mathbf{x}_1, \ldots, \mathbf{x}_n)$. Furthermore, the iterative weights $\mathbf{w}^{(l)}$ converge to $\mathbf{w}^*$ with a linear rate of convergence:*

$$\mathbf{w}^{(l)} - \mathbf{w}^* = (\mathbf{w}^{(l-1)} - \mathbf{w}^*)(\mathbf{I} - \frac{\eta}{n}\,\mathbf{X}\,\mathbf{X}^\top).$$

That is, Proposition 3 holds for the multi-layer prefixLM, so that the same exact $\mathbf{w}^*$ is also the stationary point of prefixLM, to which it converges in a linear rate. Furthermore this stationary point is exactly the (optimal) least square solution of the linear regression problem.

### 4.2 CONVERGENCE OF THE CAUSALLM ICL

Following the update rule of (10), we can view a multi-layer causalLM as implicitly maintaining different weight vectors $\mathbf{w}_j$ for each position $j$. In what follows, we show that: (a) Each such position $j$ has its own stationary point $\mathbf{w}_j^*$, which appears to be different from the global optimal point $\mathbf{w}^*$ of linear regression; (b) even when the number of in-context samples $n$ grows to infinity, convergence to $\mathbf{w}^*$ is not guaranteed.

Specifically, in Appendix B we provide a proof for the following proposition:

**Proposition 4** *If $\mathbf{w}_j^{(l)} = \sum_{i=1}^{j} a_{i,j}^{(l)}\,\mathbf{x}_i^\top$ follows the iterative updates of* (10)*, then*

$$a_{i,j}^{(l)} = a_{i,i}^{(l)} \equiv a_i^{(l)} \quad \forall j \geq i,$$

*and there exist stationary points $\mathbf{w}_j^* = \sum_{i=1}^{j} a_i^*\,\mathbf{x}_i^\top$ (for $j \in 1, \ldots, n$) with coefficients from $\mathbf{a}^* = (a_1^*, \ldots, a_n^*)$ that satisfy $\mathbf{y} = \mathbf{a}^*\,\mathbf{T}$, where*

$$\mathbf{T} = \begin{pmatrix} \mathbf{x}_1^\top\,\mathbf{x}_1 & \mathbf{x}_1^\top\,\mathbf{x}_2 & \cdots & \mathbf{x}_1^\top\,\mathbf{x}_n \\ 0 & \mathbf{x}_2^\top\,\mathbf{x}_2 & \cdots & \mathbf{x}_2^\top\,\mathbf{x}_n \\ \vdots & \vdots & \ddots & \vdots \\ 0 & 0 & \cdots & \mathbf{x}_n^\top\,\mathbf{x}_n \end{pmatrix}.$$

*Furthermore, the coefficients $\mathbf{a}^{(l)}$ converges to the stationary point $\mathbf{a}^*$ with linear rate of convergence:*

$$\mathbf{a}^{(l)} - \mathbf{a}^* = (\mathbf{a}^{(l-1)} - \mathbf{a}^*)(\mathbf{I} - \frac{\eta}{n}\,\mathbf{T}).$$

This proposition implies that the stationary points $\mathbf{w}_j^*$ of causalLM-ICL are different from $\mathbf{w}^*$, the least square solution of linear regression. However, a natural question is: if $j$ increases, would $\mathbf{w}_j^*$ ultimately converge to the optimal solution?

To answer this question, the next proposition shows that the stationary points $\mathbf{w}_j^*$ follow an online gradient descent algorithm, whose loss and gradient at the $j$-th step is,

$$L_j(\mathbf{w}_j) = \frac{1}{2}(\mathbf{w}_j\,\mathbf{x}_{j+1} - y_{j+1})^2,$$

$$\nabla_{\mathbf{w}_j}L_j(\mathbf{w}_j) = (\mathbf{w}_j\,\mathbf{x}_{j+1} - y_{j+1})\,\mathbf{x}_{j+1}^\top.$$

**Proposition 5** *Assuming that $\mathbf{w}_j^*$ is the stationary points obtained in Proposition 4, then*

$$\mathbf{w}_{j+1}^* = \mathbf{w}_j^* - \frac{1}{\|\,\mathbf{x}_{j+1}\,\|_2^2}\nabla_{\mathbf{w}_j^*}L_j(\mathbf{w}_j^*). \tag{11}$$

The proof of Proposition 5 is provided in Appendix B. Note that online gradient descent is known to converge to an optimal solution only with a decaying step size $j^{-\nu}$ for $\nu > 0$ (Jentzen & Von Wurstemberger, 2020). Since the step size of (11) does not decay, we conclude that causalLM may not converge to $\mathbf{w}^*$ even with increasing layers and increasing number of in-context examples. More concretely, as for the case of in-context learning, where the number of in-context examples $n$ is limited, convergence to the optimal solution $\mathbf{w}^*$ cannot be achieved by causalLM-ICL.

## 5 NUMERICAL EXPERIMENTS

Our experiments contain three parts.

- We first use LSA-transformers on linear regression problems to directly verify our theorems. In Section 5.1, we show that despite that the in-context example (training) error of causalLM and prefixLM both decays in linear rates, the query (test) error of causalLM is significantly larger, which indicates its stationary solution is not optimal.

- Secondly, we use ordinary softmax transformers on a few synthetic tasks including linear regression, nonlinear regression and multiclass classification. In Section 5.2, we show that our theoretical insights generalize to other tasks types (i.e., that ICL prefixLM still outperforms causalLM in all these cases).

- Lastly, in Section 5.3, we conduct LLM based ICL experiments using T5 (Roberts et al., 2022). We also provide additional experimental results on PaLM2 (Google et al., 2023) as well as large multimodal models (PaLI-X, Chen et al. (2023)) in Appendix E.6 and E.7.

### 5.1 LSA-TRANSFORMERS ON LINEAR REGRESSION

In order to directly verify our theorems from Section 4, we first study in-context learning on linear regression problem with the LSA transformer of (5). Each of the input sequence contains 40 in-context examples and 200 queries, and each query attends to all the in-context examples but does not attend to each other. See Appendix E for an illustration. The data input $\mathbf{x}_i$ of the sequence is sampled from $\mathcal{U}(-1, 1)^{16}$. Each sequence is associated with a single weight vector $\mathbf{w}$ that is sampled from $\mathcal{N}(0, \mathbf{I})$, and the labels are computed as $y_i = \mathbf{w}\,\mathbf{x}_i$. Assuming the prediction of each layer is $\tilde{y}_i^{(l)}$, we evaluate the MSE $\|y_i - \tilde{y}_i^{(l)}\|_2^2$ on both in-context and query examples across different layers $l$.

The results are plotted in Figure 2 left (for prefixLM) and middle (for causalLM). Our results are averaged over 64 randomly generated sequences. As we can see, although both prefixLM and causalLM has a linear rate of convergence (with respect to the number of layers) on the in-context examples, the query errors of causalLM are stuck above the $10^{-1}$ level, while the query error of prefixLM decays in the same linear rate as its training error.

Furthermore, in Figure 2 right, we plot the query errors of the stationary points (following Proposition 4, corresponding to the outputs of infinite layers) of causalLM-ICL with increasing number of in-context examples up to 300. Although causalLM-ICL is able to eventually converge to optimal solution when $\mu_x = 0$, it takes more than 100 examples to reach below $10^{-2}$. The convergence is

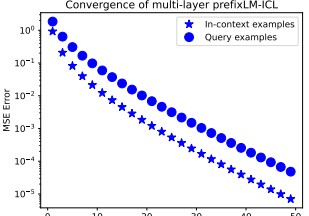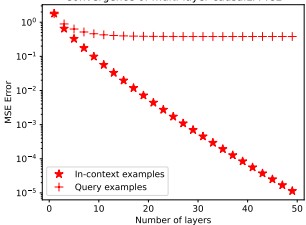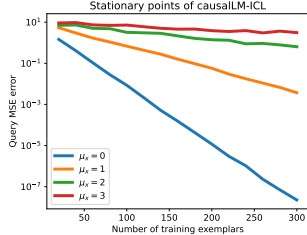

Figure 2: Left/Middle: the MSE on in-context examples and query examples of multi-layer LSA-based prefixLM/causalLM-ICLs with 40 in-context training examples. Right: the query MSE of causalLM-ICL's stationary points (per Proposition 4) using up to 300 in-context examples.

|            | LR     | N-LR   | MC    |
|------------|--------|--------|-------|
| PrefixLM-SL | 8.6e-3 | 1.5e-4 | **24.1** |
| CausalLM-SL | 1.9e-1 | 2.7e-3 | 27.0  |
| PrefixLM-UL | **2.5e-3** | **9.0e-5** | 27.6  |
| CausalLM-UL | 1.6e-2 | 2.9e-3 | 32.1  |

Table 1: The test query errors of the unshared-layer (UL) and sharing-layer (SL) transformer-ICLs on linear regression (LR), non-linear regression (NLR), and multiclass classification (MC) tasks. Both regression tasks report mean squared errors; and the MC task reports the classification error.

even worse as we vary the input distribution $\mathbf{x} \sim \mathcal{U}(-1,1)^d + \mu_x$ with increasing $\mu_x \in \{0,1,2,3\}$, which demonstrates that causalLM-ICL is not optimal for few-shot ICL.

## 5.2 STANDARD TRANSFORMERS ON SYNTHETIC TASKS

Previous experiments provided a proof of concept verification of the propositions from Section 4. Next we verify if a standard softmax transformer-based prefixLM and causalLM ICL exhibit similar differences on various types of synthetic tasks including linear regression, non-linear regression and multiclass classification.

All three tasks used 16-dim inputs with $\mathbf{x} \sim \mathcal{U}(-1,1)^{16}$ and $\mathbf{w} \sim \mathcal{N}(0,\mathbf{I})$. For non-linear regression, we apply a sigmoid activation on the logit such that $y = \mathrm{sigmoid}(\mathbf{w}\,\mathbf{x})$; and for multiclass classification, we randomly generate three $\mathbf{w}_c \sim \mathcal{N}(0,\mathbf{I})$, and assign labels based on $y = \mathrm{argmax}_c\{\mathbf{w}_c\,\mathbf{x}\}$. We trained a few 24-layer transformers containing 128 hidden units and 2 heads. Besides of the comparisons of prefixLM and causalLM, we also compare the transformers with or without sharing layers (SL vs UL). In particular, the sharing-layer transformer can be considered a recurrent system (Dehghani et al., 2018) where the dynamic is continual along the layers and a stationary point may exist given infinite number of layers, which makes it closer to our constructed LSA.

The ICL training dataset contains 64,000 training sequences. Each sequence contains 40 in-context examples and 20 queries, where queries are independent of each other similar to Section 5.1. The transformers are trained with batch size 64 for 100 epochs. More details of the hyper-parameters of the experiments are provided in Appendix E.

We evaluate the ICL performance using 64 holdout test sequences and report the test errors on the query examples. The results are summarized in Table 1. We find that both prefixLM-SL and prefixLM-UL significantly outperform their counterparts of causalLM in all cases. As a side note, transformer-SL appears to outperform transformer-UL in the classification tasks, which indicates the overfitting problem of the latter due to over-parameterization. In addition, we also add probes at the output of each SL-transformer layer to visualize the test errors of intermediate layers in Figure 3. Comparing Figure 3 and Figure 2 (left/middle) reveals some similarities. Although the test query errors of causalLM decay in roughly the same rate as the ones of prefixLM in earlier layers, the decays become much slower in later layers possibly due to the nature of its non-optimal stationary points. These results suggest that the title argument of the paper also holds beyond LSA-based transformers and linear regression.

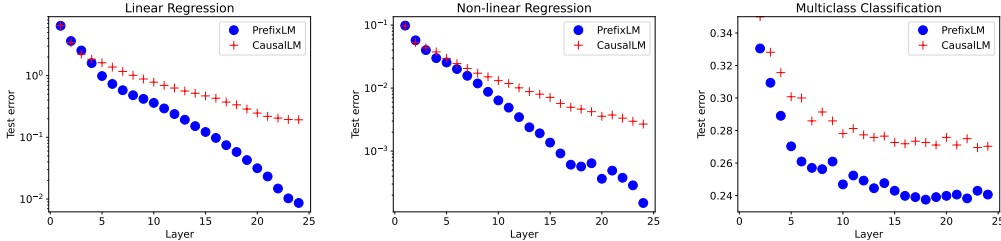

Figure 3: The test query errors of the 24-layer SL-transformers based prefixLM/causalLM-ICLs on linear regression (left), non-linear regression (middle), and multiclass classification (right).

| | MMLU | | | BBH | | |
|---|---|---|---|---|---|---|
| | Base | Large | XL | Base | Large | XL |
| PrefixLM | **28.8** | **32.0** | **39.5** | **27.4** | **32.2** | **35.8** |
| CausalLM | 28.0 | 26.9 | 30.5 | 24.8 | 29.8 | 32.0 |

Table 2: The averaged test query accuracies on 5-shot MMLU (57 tasks) and 3-shot BBH (23 tasks) with FLAN-finetuned T5 DecoderOnly prefixLM/causalLM checkpoints.

## 5.3 ICL ON LARGE LANGUAGE MODELS

In order to compare the ICL performance of causalLM and prefixLM in a large language model setting, we conduct experiments using the publicly available T5 family of models (Roberts et al., 2022). Note that the existing public T5X [§] checkpoints are all based on EncDec models, which are similar to prefixLM. Thus, it would be unfair and unnatural to compare with causalLM by simply replacing the bidirectional attention of the encoder to the causal attention during the finetuning stage. To make a more reasonable comparison, we reran the pretraining stages of T5 on the C4 corpus (Raffel et al., 2020a) from a random initialization point using a span corruption objective, but in the DecoderOnly setting. Moreover, for each size (from Base, Large and XL) of the models, we pretrained two checkpoints, one with prefixLM and the other with causalLM, each for 1M steps using the same T5 pretraining recipe. After pretraining, we use the FLAN recipe (Chung et al., 2022) to finetune each checkpoint (40k steps for Base, 20k steps for Large and XL) with its pretrained attention mask and evaluate the ICL capability of the finetuned models on two benchmarks: MMLU (Hendrycks et al., 2020) and BBH (Suzgun et al., 2022).

Table 2 shows that for all three sizes of T5X DecoderOnly models, the MMLU and BBH accuracies of prefixLM surpasses that of causalLM consistently and such gap widens as the size of the model becomes larger. This result empirically verifies that our conjecture generalizes to the practical case. We supply additional empirical evidence on state-of-the-art models in Appendix E.6 and E.7.

## 6 CONCLUSION

In this paper, we analyzed the convergence properties of two types of widely-used transformer-based language models (causalLM and prefixLM), during in-context learning. Using a simplified LSA attention in a linear regression setting, we proved that both LM types converge to their stationary points in linear rates, but that their stationary points have significantly different properties. In particular, the stationary points of prefixLM coincides with the optimal least square solution; while the ones of causalLM is equivalent to the weights of an online learning system, that is not guaranteed to converge to the optimal solution. Our experiments verified the above theoretical results, and also empirically extend the findings to general transformer on non-linear regression as well as classification tasks. Finally, we compare causalLM and prefixLM on a few large language models and find that prefixLM also consistently wins over causalLM in practical few-shot tasks.

---

[§]https://github.com/google-research/t5x

ACKNOWLEDGEMENTS

We want to specially thank Xinhua Zhang for helpful discussions and references about online learning. We also thank Yi Tay for comments regarding the PaLM2 checkpoints.

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
