# CausalLM is not optimal for in-context learning

## Appendices

## A   RELATED WORK

Ever since GPT-3 (Brown et al., 2020) exhibited its in-context learning abilities in various language inference and translation tasks, there has been tremendous interest in understanding the mechanics behind In-Context Learning (ICL). Currently, there are two main camps of thought that try to explain ICL: (1) the representation camp, which views ICL behavior as a topic model that extracts relevant memories based on the topic of the context (Xie et al., 2021; Min et al., 2022) - these works support this view with the findings that in-context learner sometimes behaved similarly even when the label of the training examples were permuted (Min et al., 2022). (2) the algorithmic camp, which holds that LLMs learns to implement a learning algorithm (Garg et al., 2022; Akyürek et al., 2022; Von Oswald et al., 2023) and then run it during ICL - these works usually propose a construction of the transformer parameters and show that it can solve certain simple tasks (e.g. linear regression), then empirically verify that transformers track the behavior of the algorithm of interest.

Moreover, recent studies of large-scale data and model (Wei et al., 2023) discovered that large language models seem to exhibit certain emergent behavior, where, ICL is memory-based on small-to-medium sized models or data, but becomes more algorithm-based on larger model and data. For example, (Wei et al., 2023) showed that a large language model is able to respond accordingly to the flipped label in in-context examples, opposing the findings of (Min et al., 2022).

Since most ICL applications only involve few shots of context examples, it seems reasonable to conjecture that the memory of a *deep* representation and a *shallow* predictor algorithm may co-exist in contributing the in-context learning capabilities. Since the representation learning of large language models have been universally acknowledged, it is more interesting to investigate how transformer learns to in-context learn shallow predictors with few-shot examples.

Focusing on work from the algorithmic camp, we note that (Garg et al., 2022) were the first to suggest using linear regression to study in-context learning. The authors empirically found that a 12-layer transformer is able to achieve similar results as a least-square solver on a 20-dim linear regression problem with around 20 in-context examples. Beyond linear regression, they also found that transformers can in-context learn a few other classes of shallow predictors, including two-layer Relu networks.

Probably the first formal theoretical investigation of the linear regression in-context learners is (Akyürek et al., 2022). They first showed that a transformer layer can approximately conduct four basic operations: mov, mul, div, aff. They then cleverly combined these four operations and showed that a gradient descent step of linear regression can be implemented with a 4-head 8-layer transformer with $O(d)$ hidden units, where $d$ is the dimension of the inputs $\mathbf{x}$. Despite their novel construction, the result itself provides only a loose upper bound on the model size (or depth) that is required for simulating linear regression within a transformer - for example, (Von Oswald et al., 2023) reported that a 2 or 5-layer transformer already achieves significantly better results than a single-step gradient descent for linear regression.

Because of the significant discrepancy between the construction of (Akyürek et al., 2022) and the empirical results, the one-layer LSA construction of (Von Oswald et al., 2023) appears to be more appealing and matches the experimental results better. Moreover, a most recent work by (Zhang et al., 2023) used gradient flow to prove that by initializing $\mathbf{w}^{(0)} = 0$, such matrix constructions can indeed be learned by an LSA transformer. This is why our paper follows this construction and studies its multi-layer convergence properties with different types of attention (prefixLM vs causalLM).

In terms of the comparison between prefixLM and causalLM, such research work can be traced back as early as (Raffel et al., 2020b), where they showed prefixLM outperforms causalLM in varieties of NL tasks. Later, UL-2 (Tay et al., 2022) proposed to mix prefixLM and span corruption objectives, and found it to be more efficient than the causalLM objective alone. It was also shown

in (Chung et al., 2022), that U-PaLM (a UL2-finetuning PaLM) outperforms PaLM (causalLM only) in various ICL tasks. Indeed, for the reasons above, some of the latest models have included prefixLM objectives in the pretraining mix (for example PaLM-2 by Google et al. (2023)). On the other hand, prominent models such as Flamingo as well as the ones in the GPT-family are still based on the causalLM structure, so the comparison between prefixLM and causalLM remains important and relevant. Furthermore, all previous studies were done in an empirical manner, whereas we set out to explain their differences from a theoretical perspective and back the theory with empirical evidence. While we are not the first to follow this path, our work is the first to provide a theoretical justification for the advantage of prefixLM over causalLM in a multi-layer transformer ICL setting by analyzing their theoretical convergence properties.

## B  PROOFS

In this section, we provide proofs of the propositions introduced in Section 3 and Section 4.

**Proposition 1**  For a multi-layer LSA satisfying (5) with $\mathbf{w}^{(0)} = 0$, if its input $\mathbf{Z}$ is formatted as (4), then its $l$-th layer output is $\mathbf{z}_j^{(l)} = (\mathbf{x}_j^\top, \delta_j^{(l)})^\top$, where $\delta_j^{(l)} = y_j - \mathbf{w}^{(l)} \mathbf{x}_j$ and $\mathbf{w}^{(l)}$ is the weight from the $l$-th step gradient descents as in (3).

**Proof:**  Plugging in $\mathbf{K}$, $\mathbf{Q}$, $\mathbf{P}$ and $\mathbf{V}$ of (5) with $\mathbf{w}^{(0)} = 0$ into (1), we have

$$\begin{pmatrix} \mathbf{x}_j \\ \delta_j^{(l)} \end{pmatrix} = \begin{pmatrix} \mathbf{x}_j \\ \delta_j^{(l-1)} \end{pmatrix} + \frac{\eta}{n} \begin{pmatrix} \mathbf{0}_{d\times d} & \mathbf{0} \\ 0 & -1 \end{pmatrix} \cdot$$

$$\left( \sum_{i=1}^n \begin{pmatrix} \mathbf{x}_i \\ \delta_i^{(l-1)} \end{pmatrix} (\mathbf{x}_i^\top, \delta_i^{(l-1)}) \begin{pmatrix} \mathbf{I}_{d\times d} & \mathbf{0} \\ 0 & 0 \end{pmatrix} \begin{pmatrix} \mathbf{x}_j \\ \delta_j^{(l-1)} \end{pmatrix} \right)$$

$$= \begin{pmatrix} \mathbf{x}_j \\ \delta_j^{(l-1)} \end{pmatrix} - \frac{\eta}{n} \sum_{i=1}^n \begin{pmatrix} \mathbf{0} \\ \delta_i^{(l-1)} \end{pmatrix} \mathbf{x}_i^\top \mathbf{x}_j .$$

It is easy to see that $\mathbf{z}_j$ never changes its first $d$-dimension corresponding to $\mathbf{x}_j$. Therefore, we can simplify the above equation and focus only on the last coordinate $\delta_j^{(l)}$, where we have

$$\delta_j^{(l)} = \delta_j^{(l-1)} - \frac{\eta}{n} \sum_{i=1}^n \delta_i^{(l-1)} \mathbf{x}_i^\top \mathbf{x}_j, \tag{12}$$

with $\delta_j^{(0)} = y_j$. Defining $\tilde{y}_j^{(l)} = y_j - \delta_j^{(l)}$ and rearranging (12), we obtain $\tilde{y}_j^{(0)} = 0$ and

$$\tilde{y}_j^{(l)} = \tilde{y}_j^{(l-1)} + \frac{\eta}{n} \sum_{i=1}^n (y_i - \tilde{y}_i^{(l-1)}) \mathbf{x}_i^\top \mathbf{x}_j . \tag{13}$$

Next we prove $\tilde{y}_j^{(l)} = \mathbf{w}^{(l)} \mathbf{x}_j$ by induction. Since $\mathbf{w}^{(0)} = 0$, it is clear that $\tilde{y}_j^{(0)} = \mathbf{w}^{(0)} \mathbf{x}_j = 0$ for all $j$.

If $\tilde{y}_j^{(l-1)} = \mathbf{w}^{(l-1)} \mathbf{x}_j$ for all $j$, then

$$\tilde{y}_j^{(l)} = \tilde{y}_j^{(l-1)} + \frac{\eta}{n} \sum_{i=1}^n (y_i - \tilde{y}_i^{(l-1)}) \mathbf{x}_i^\top \mathbf{x}_j$$

$$= \mathbf{w}^{(l-1)} \mathbf{x}_j + \frac{\eta}{n} \sum_{i=1}^n (y_i - \mathbf{w}^{(l-1)} \mathbf{x}_i) \mathbf{x}_i^\top \mathbf{x}_j$$

$$= \left( \mathbf{w}^{(l-1)} + \sum_{i=1}^n (y_i - \mathbf{w}^{(l-1)} \mathbf{x}_i) \mathbf{x}_i^\top \right) \mathbf{x}_j$$

$$= \mathbf{w}^{(l)} \mathbf{x}_j .$$

$\square$

**Proposition 2** For a multi-layer causalLM-LSA satisfying (5) with $\mathbf{w}^{(0)} = 0$, if its input $\mathbf{Z}$ is formatted as (4), then its $l$-th layer output is $\mathbf{z}_j^{(l)} = (\mathbf{x}_j^\top, \delta_j^{(l)})^\top$, where $\delta_j^{(l)} = y_j - \mathbf{w}_j^{(l)} \mathbf{x}_j$ and $\mathbf{w}_j^{(l)}$ follow (10).

**Proof:** Plugging in $\mathbf{K}$, $\mathbf{Q}$, $\mathbf{P}$ and $\mathbf{V}$ of (5) with $\mathbf{w}^{(0)} = 0$ into (2), we have

$$\delta_j^{(l)} = \delta_j^{(l-1)} - \frac{\eta}{n} \sum_{i=1}^{j} \delta_i^{(l-1)} \mathbf{x}_i^\top \mathbf{x}_j$$

$$\tilde{y}_j^{(l)} = \tilde{y}_j^{(l-1)} + \frac{\eta}{n} \sum_{i=1}^{j} (y_i - \tilde{y}_i^{(l-1)}) \mathbf{x}_i^\top \mathbf{x}_j$$

with $\tilde{y}_j^{(l)} = y_j - \delta_j^{(l)}$. Next we prove $\tilde{y}_j^{(l)} = \mathbf{w}_j^{(l)} \mathbf{x}_j$ by induction. Since $\mathbf{w}_j^{(0)} = 0$, it is clear that $\tilde{y}_j^{(0)} = \mathbf{w}_j^{(0)} \mathbf{x}_j = 0$ for all $j$.

If $\tilde{y}_j^{(l-1)} = \mathbf{w}_j^{(l-1)} \mathbf{x}_j$ for all $j$, then

$$\tilde{y}_j^{(l)} = \tilde{y}_j^{(l-1)} + \frac{\eta}{j} \sum_{i=1}^{n} (y_i - \tilde{y}_i^{(l-1)}) \mathbf{x}_i^\top \mathbf{x}_j$$

$$= \mathbf{w}_j^{(l-1)} \mathbf{x}_j + \frac{\eta}{n} \sum_{i=1}^{j} (y_i - \mathbf{w}_i^{(l-1)} \mathbf{x}_i) \mathbf{x}_i^\top \mathbf{x}_j$$

$$= \left( \mathbf{w}_j^{(l-1)} + \sum_{i=1}^{n} (y_i - \mathbf{w}_i^{(l-1)} \mathbf{x}_i) \mathbf{x}_i^\top \right) \mathbf{x}_j$$

$$= \mathbf{w}_j^{(l)} \mathbf{x}_j .$$

$\square$

**Proposition 3** If $\mathbf{w}^{(l)}$ follows the iterative updates of (3), then there exists a stationary point $\mathbf{w}^*$ with coefficients satisfying:

$$\mathbf{y} \mathbf{x}^\top = \mathbf{w}^* \mathbf{X} \mathbf{X}^\top ,$$

where $\mathbf{y} = (y_1, \ldots, y_n)$ and $\mathbf{X} = (\mathbf{x}_1, \ldots, \mathbf{x}_n)$. Furthermore, the iterative weights $\mathbf{w}^{(l)}$ converges to the stationary point $\mathbf{w}^*$ with linear rate of convergence:

$$\mathbf{w}^{(l)} - \mathbf{w}^* = (\mathbf{w}^{(l-1)} - \mathbf{w}^*)(\mathbf{I} - \frac{\eta}{n} \mathbf{X} \mathbf{X}^\top).$$

**Proof:** From (3), we have

$$\mathbf{w}^{(l)} = \mathbf{w}^{(l-1)} + \frac{\eta}{n} \underbrace{\sum_{i=1}^{n} (y_i - \mathbf{w}^{(l-1)} \mathbf{x}_i) \mathbf{x}_i^\top}_{(*)} .$$

The stationary point must satisfy $(*) = 0$. Written in vectorized form, we have

$$\mathbf{y} \mathbf{X}^\top = \mathbf{w}^* \mathbf{X} \mathbf{X}^\top . \tag{14}$$

Now plugging (14) back to (3), we have

$$\mathbf{w}^{(l)} = \mathbf{w}^{(l-1)} + \frac{\eta}{n} \left( \mathbf{w}^* \mathbf{X} \mathbf{X}^\top - \mathbf{a} \mathbf{w}^{(l-1)} \mathbf{X} \mathbf{X}^\top \right),$$

which can be reorganized to

$$\mathbf{w}^{(l)} - \mathbf{w}^* = (\mathbf{w}^{(l-1)} - \mathbf{w}^*)(\mathbf{I} - \frac{\eta}{n} \mathbf{X} \mathbf{X}^\top).$$

$\square$

**Proposition 4** If $\mathbf{w}_j^{(l)} = \sum_{i=1}^j a_{i,j}^{(l)} \mathbf{x}_i^\top$ follows the iterative updates of (10), then

$$a_{i,j}^{(l)} = a_{i,i}^{(l)} \equiv a_i^{(l)} \quad \forall j \geq i,$$

and there exists stationary points $\mathbf{w}_j^* = \sum_{i=1}^j a_i^* \mathbf{x}_i^\top$ (for $j \in 1, \ldots, n$) with coefficients from $\mathbf{a}^* = (a_1^*, \ldots, a_n^*)$ that satisfy $\mathbf{y} = \mathbf{a}^* \, \mathbf{T}$, where

$$\mathbf{T} = \begin{pmatrix} \mathbf{x}_1^\top \mathbf{x}_1 & \mathbf{x}_1^\top \mathbf{x}_2 & \cdots & \mathbf{x}_1^\top \mathbf{x}_n \\ 0 & \mathbf{x}_2^\top \mathbf{x}_2 & \cdots & \mathbf{x}_2^\top \mathbf{x}_n \\ \vdots & \vdots & \ddots & \vdots \\ 0 & 0 & \cdots & \mathbf{x}_n^\top \mathbf{x}_n \end{pmatrix}.$$

Furthermore, the coefficients $\mathbf{a}^{(l)}$ converges to the stationary point $\mathbf{a}^*$ with linear rate of convergence:

$$\mathbf{a}^{(l)} - \mathbf{a}^* = (\mathbf{a}^{(l-1)} - \mathbf{a}^*)(\mathbf{I} - \frac{\eta}{n}\,\mathbf{T}).$$

**Proof:** First notice that according to (10), we have

$$\mathbf{w}_j^{(l)} = \mathbf{w}_j^{(l-1)} + \frac{\eta}{n} \sum_{i=1}^j (y_i - \mathbf{w}_i^{(l-1)}\mathbf{x}_i)\mathbf{x}_i^\top$$

$$= \sum_{i=1}^j \left(a_{i,j}^{(l-1)} + \frac{\eta}{n}(y_i - \mathbf{w}_i^{(l-1)}\mathbf{x}_i)\right)\mathbf{x}_i^\top$$

or

$$a_{i,j}^{(l)} = a_{i,j}^{(l-1)} + \frac{\eta}{n}(y_i - \mathbf{w}_i^{(l-1)}\mathbf{x}_i) \quad \forall j \geq i.$$

Since $a_{i,j}^{(0)} = 0$, and the above update is the same for any $j$ given any $i$, then it is obvious by induction that

$$a_{i,j}^{(l)} = a_{i,i}^{(l)} \equiv a_i^{(l)} \quad \forall j \geq i.$$

Therefore, we can simplify $\mathbf{w}_j^{(l)} = \sum_{i=1}^j a_i^{(l)} \mathbf{x}_i^\top$.

Now plugging into (10), we have

$$\sum_{i=1}^j a_i^{(l)} \mathbf{x}_i^\top$$

$$= \sum_{i=1}^j a_i^{(l-1)} \mathbf{x}_i^\top + \frac{\eta}{n} \sum_{i=1}^j (y_i - \sum_{k=1}^i a_k^{(l-1)} \mathbf{x}_k^\top \mathbf{x}_i)\mathbf{x}_i^\top$$

$$= \sum_{i=1}^j \left(a_i^{(l-1)} + y_i - \frac{\eta}{n}\left(\sum_{k=1}^i a_k^{(l-1)} \mathbf{x}_k^\top \mathbf{x}_i\right)\right)\mathbf{x}_i^\top,$$

which is equivalent to

$$a_i^{(l)} = a_i^{(l-1)} + \frac{\eta}{n}\underbrace{\left(y_i - \sum_{k=1}^i a_k^{(l-1)} \mathbf{x}_k^\top \mathbf{x}_i\right)}_{(*)}. \tag{15}$$

The stationary points satisfy $(*) = 0$, which gives

$$y_1 = a_1^* \mathbf{x}_1^\top \mathbf{x}_1$$
$$y_2 = a_1^* \mathbf{x}_1^\top \mathbf{x}_2 + a_2^* \mathbf{x}_2^\top \mathbf{x}_2$$
$$\cdots$$
$$y_n = a_1^* \mathbf{x}_1^\top \mathbf{x}_n + \ldots + a_n^* \mathbf{x}_n^\top \mathbf{x}_n,$$

or in the vectorized form $\mathbf{y} = \mathbf{a}^* \, \mathbf{T}$, where

$$\mathbf{T} = \begin{pmatrix} \mathbf{x}_1^\top \mathbf{x}_1 & \mathbf{x}_1^\top \mathbf{x}_2 & \cdots & \mathbf{x}_1^\top \mathbf{x}_n \\ 0 & \mathbf{x}_2^\top \mathbf{x}_2 & \cdots & \mathbf{x}_2^\top \mathbf{x}_n \\ \vdots & \vdots & \ddots & \vdots \\ 0 & 0 & \cdots & \mathbf{x}_n^\top \mathbf{x}_n \end{pmatrix}.$$

Now plugging in $\mathbf{y} = \mathbf{a}^* \, \mathbf{T}$ back to (15) and vectorize it, yields

$$\mathbf{a}^{(l)} = \mathbf{a}^{(l-1)} + \frac{\eta}{n} \left( \mathbf{a}^* \, \mathbf{T} - \mathbf{a} \, \mathbf{T} \right),$$

which can be reorganized to

$$\mathbf{a}^{(l)} - \mathbf{a}^* = (\mathbf{a}^{(l-1)} - \mathbf{a}^*)(\mathbf{I} - \frac{\eta}{n} \, \mathbf{T}).$$

$\square$

**Proposition 5**   Assuming that $\mathbf{w}_j^*$ is the stationary points obtained in Proposition 4, then

$$\mathbf{w}_{j+1}^* = \mathbf{w}_j^* - \frac{1}{\| \, \mathbf{x}_{j+1} \, \|_2^2} \nabla_{\mathbf{w}_j^*} L_j(\mathbf{w}_j^*).$$

**Proof:**   Recall the online learning system with a sequence of data-label pairs $(\mathbf{x}_j, y_j)$ has the following online loss and its gradient at the $j$-th step,

$$L_j(\mathbf{w}_j) = \frac{1}{2}(\mathbf{w}_j \, \mathbf{x}_{j+1} - y_{j+1})^2,$$

$$\nabla_{\mathbf{w}_j} L_j(\mathbf{w}_j) = (\mathbf{w}_j \, \mathbf{x}_{j+1} - y_{j+1}) \, \mathbf{x}_{j+1}^\top.$$

According to Proposition 4, we have $\mathbf{y} = \mathbf{a}^* \, \mathbf{T}$, which gives

$$\begin{aligned} y_{j+1} =& a_1^* \mathbf{x}_1^\top \mathbf{x}_{j+1} + \ldots + a_j^* \mathbf{x}_j^\top \mathbf{x}_{j+1} \\ & + a_{j+1}^* \mathbf{x}_{j+1}^\top \mathbf{x}_{j+1} \\ =& \mathbf{w}_j^* \, \mathbf{x}_{j+1} + a_{j+1}^* \mathbf{x}_{j+1}^\top \mathbf{x}_{j+1} \end{aligned} \qquad (16)$$

where the last equation is due to $\mathbf{w}_j^* = \sum_{i=1}^j a_i^* \, \mathbf{x}_i^\top$.

Since $\mathbf{w}_j^* = \sum_{i=1}^j a_i^* \, \mathbf{x}_i^\top$, we have

$$\begin{aligned} \mathbf{w}_{j+1}^* =& \mathbf{w}_j^* + a_{j+1}^* \mathbf{x}_{j+1}^\top \\ =& \mathbf{w}_j^* + \frac{1}{\| \, \mathbf{x}_{j+1} \, \|_2^2} \left( a_{j+1}^* \mathbf{x}_{j+1}^\top \mathbf{x}_{j+1} \right) \mathbf{x}_{j+1}^\top \\ =& \mathbf{w}_j^* - \frac{1}{\| \, \mathbf{x}_{j+1} \, \|_2^2} (\mathbf{w}_j^* \, \mathbf{x}_{j+1} - y_{j+1}) \, \mathbf{x}_{j+1}^\top \\ =& \mathbf{w}_j^* - \frac{1}{\| \, \mathbf{x}_{j+1} \, \|_2^2} \nabla_{\mathbf{w}_j^*} L_j(\mathbf{w}_j^*) \end{aligned}$$

where the third equation is because of (16).

$\square$

## C   MULTI-LAYER LSA CONSTRUCTION WITH NON-ZERO W(0)

In this section, we introduce the proposition that connects a multi-layer LSA following the construction of (5) but with non-zero $\mathbf{w}^{(0)}$ and the multi-step gradient descents of linear regression.

**Proposition 6**   *For a multi-layer LSA satisfying the construction* (5)*, if its input* $\mathbf{Z}$ *is formatted as* (4)*, then its $l$-th layer output is* $\mathbf{z}_j^{(l)} = (\mathbf{x}_j^\top, \delta_j^{(l)})^\top$*, where* $\delta_j^{(l)} = y_j - (\mathbf{w}^{(l)} - \mathbf{w}^{(0)}) \, \mathbf{x}_j$ *and* $\mathbf{w}^{(l)}$ *is the $l$-th updated weight from the gradient descents update rule in* (3)*.*

**Proof:** Plugging in $\mathbf{K}$, $\mathbf{Q}$, $\mathbf{P}$ and $\mathbf{V}$ of (5) into (1), we have

$$\delta_j^{(l)} = \delta_j^{(l-1)} - \frac{\eta}{n} \sum_{i=1}^{n} \left( \delta_i^{(l-1)} - \mathbf{w}^{(0)} \mathbf{x}_i \right) \mathbf{x}_i^\top \mathbf{x}_j, \tag{17}$$

with $\delta_j^{(0)} = y_j$. Defining $\tilde{y}_j^{(l)} = y_j - \delta_j^{(l)} + \mathbf{w}^{(0)} \mathbf{x}_j$ and rearranging the (17), we obtain $\tilde{y}_j^{(0)} = 0$ and

$$\tilde{y}_j^{(l)} = \tilde{y}_j^{(l-1)} + \frac{\eta}{n} \sum_{i=1}^{n} (y_i - \tilde{y}_i^{(l-1)}) \mathbf{x}_i^\top \mathbf{x}_j \,.$$

Then it is easy to prove $\tilde{y}_j^{(l)} = \mathbf{w}^{(l)} \mathbf{x}_j$ by induction, similar to the proof of Proposition 1. $\qquad\square$

## D CAUSALLM WITH ATTENTION-LENGTH-BASED COEFFICIENTS

Since there are $j$ terms in the summation of (10), another reasonable update for causalLM would be

$$\mathbf{w}_j^{(l)} = \mathbf{w}_j^{(l-1)} + \frac{\eta}{j} \sum_{i=1}^{j} (y_i - \mathbf{w}_i^{(l-1)} \mathbf{x}_i) \mathbf{x}_i^\top, \tag{18}$$

which we call causalLM2. For causalLM2, we have the following proposition.

**Proposition 7** *If $\mathbf{w}_j^{(l)} = \sum_{i=1}^{j} a_{i,j}^{(l)} \mathbf{x}_i^\top$ follows the iterative updates of (18), then*

$$a_{i,j}^{(l)} \equiv \frac{1}{j} a_i^{(l)} \quad \forall j \geq i,$$

*and there exists stationary points $\mathbf{w}_j^* = \frac{1}{j} \sum_{i=1}^{j} a_i^* \mathbf{x}_i^\top$ (for $j \in 1, \ldots, n$) with coefficients from $\mathbf{a}^* = (a_1^*, \ldots, a_n^*)$ that satisfy $\mathbf{y} = \mathbf{a}^* \mathbf{S}$, where*

$$\mathbf{S} = \begin{pmatrix} \mathbf{x}_1^\top \mathbf{x}_1 & \frac{1}{2} \mathbf{x}_1^\top \mathbf{x}_2 & \cdots & \frac{1}{n} \mathbf{x}_1^\top \mathbf{x}_n \\ 0 & \frac{1}{2} \mathbf{x}_2^\top \mathbf{x}_2 & \cdots & \frac{1}{n} \mathbf{x}_2^\top \mathbf{x}_n \\ \vdots & \vdots & \ddots & \vdots \\ 0 & 0 & \cdots & \frac{1}{n} \mathbf{x}_n^\top \mathbf{x}_n \end{pmatrix}.$$

*Furthermore, the coefficients $\mathbf{a}^{(l)}$ converges to the stationary point $\mathbf{a}^*$ with the following rate of convergence:*

$$\mathbf{a}^{(l)} - \mathbf{a}^* = (\mathbf{a}^{(l-1)} - \mathbf{a}^*)(\mathbf{I} - \eta \mathbf{S}).$$

The condition number $\kappa(\mathbf{S})$ is about $n/2$ greater than the one of $\kappa(\mathbf{T})$, which makes causalLM2 converge much slower than causalLM.

One can also prove that the stationary point of causalLM2 corresponds to the following online system with online loss and gradient at the $j$-th step,

$$L_j(\tilde{\mathbf{w}}_j) = \frac{1}{2} (\tilde{\mathbf{w}}_j \mathbf{x}_{j+1} - y_{j+1})^2,$$

$$\nabla_{\tilde{\mathbf{w}}_j} L_j(\tilde{\mathbf{w}}_j) = (\tilde{\mathbf{w}}_j \mathbf{x}_{j+1} - y_{j+1}) \mathbf{x}_{j+1}^\top,$$

where $\tilde{\mathbf{w}} = \frac{j}{j+1} \mathbf{w}$.

**Proposition 8** *Assuming that $\mathbf{w}_j^*$ is the stationary points obtained in Proposition 4, then*

$$\mathbf{w}_{j+1}^* = \tilde{\mathbf{w}}_j^* - \frac{1}{\| \mathbf{x}_{j+1} \|_2^2} \nabla_{\tilde{\mathbf{w}}_j^*} L_j(\tilde{\mathbf{w}}_j^*).$$

Since the step does not have $j^{-\nu}$ ($\nu > 0$) decay, such online system is not guaranteed to converge, therefore suffers the same problem as the original causalLM in Section 3.2.

In Figure 4, we plot the query MSE error of the stationary points of causalLM2-ICL with increasing number of in-context examples. We can see that the online system corresponding to causalLM2-ICL converges even slower than the ones of causalLM-ICL in Figure 2 right.

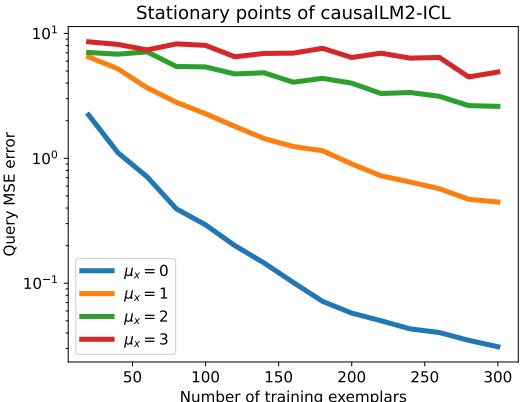

Figure 4: The test error on the stationary point of the causalLM2-ICL with up to 300 in-context examples.

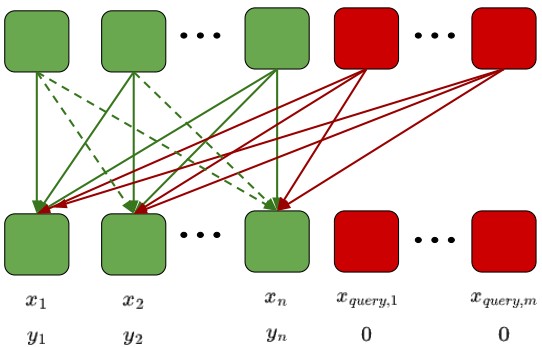

Figure 5: The illustration of the attention mask. Green arrows represent the attentions between in-context examples. The dashed arrows only applies for prefixLM. Red arrows represent the attentions from queries to in-context examples. The query examples should not attend to themselves because the inputs do not contain labels.

# E  ADDITIONAL EXPERIMENTAL DETAILS AND RESULTS

## E.1  EXPERIMENT SETTINGS FOR SECTION 5.1

In order to directly verify the theorem, we used the constructed LSA-based transformer, with $\mathbf{K} = \mathbf{Q} = \begin{pmatrix} \mathbf{I}_{d \times d} & \mathbf{0} \\ 0 & 0 \end{pmatrix}$, $\mathbf{V} = \begin{pmatrix} \mathbf{0}_{d \times d} & \mathbf{0} \\ 0 & -1 \end{pmatrix}$ and $\mathbf{P} = \frac{\eta}{n} \mathbf{I}$. Although not a trained transformer, it was recently proved in (Zhang et al., 2023) that a randomly initialized LSA-transformer does converge to such a construction. In addition, we did an ablation test of $\eta = \{0.1, 0.2, 0.4, 0.8, 1.6, 3.2\}$ and chose $\eta = 1.6$ as it converges the fastest without any divergence problems.

We randomly generated 64 sequences for ICL evaluation. For each sequence, we put the first 40 examples as the in-context examples and the last 200 examples as the query examples. The queries are independent of each other without attention. See Figure 5 for an illustration of the transformer attention mask. Such multi-query design is for training efficiency purpose only and is equivalent to 200 sequences with the same $\mathbf{w}$ and input examples $\mathbf{x}_i$, but different one query per sequence.

## E.2  EXPERIMENT SETTINGS FOR SECTION 5.2

In order to verify that our theorems can be qualitatively applied beyond LSA and linear regression, we conducted several experiments over various synthetic tasks using regular transformers. We based our

|  | LR | N-LR | MC |
|---|---|---|---|
| PrefixLM-SL-L2 | 8.6e-3 | 1.5e-4 | 24.1 |
| CausalLM-SL-L2 | 1.9e-1 | 2.7e-3 | 27.0 |
| PrefixLM-SL-no-L2 | 6.7e-3 | 1.5e-4 | 24.5 |
| CausalLM-SL-no-L2 | 5.0e-2 | 1.9e-3 | 30.5 |
| PrefixLM-UL-L2 | 7.6e-3 | 1.7e-4 | 26.7 |
| CausalLM-UL-L2 | 4.4e-2 | 2.5e-3 | 30.4 |
| PrefixLM-UL-no-L2 | 2.5e-3 | 9.0e-5 | 27.6 |
| CausalLM-UL-no-L2 | 1.6e-2 | 2.9e-3 | 32.1 |

Table 3: The test query errors of the unshared-layer (UL) and sharing-layer (SL) transformer-ICLs with or without L2 regularizer on linear regression (LR), non-linear regression (NLR), and multiclass classification (MC) tasks.

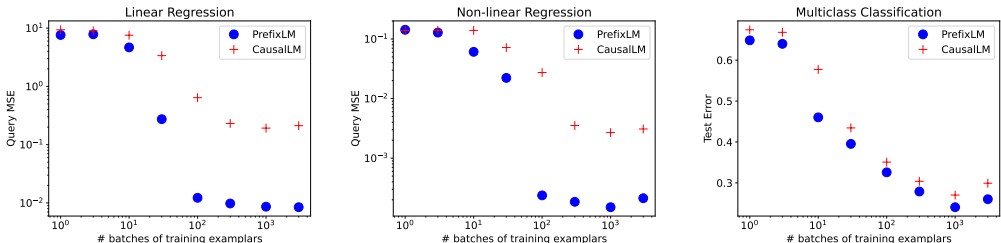

Figure 6: The test query errors of the SL-transformers based prefixLM/causalLM-ICLs with various numbers of training sequences on linear regression (left), non-linear regression (middle), and multiclass classification (right).

code from the repository of (Akyürek et al., 2022)[¶] and applied their default training hyperparameters of the code. We used a transformer of 24 layers with 128 hidden units and 2 heads. The FFN intermediate size is $4 \times 128 = 512$. The learning schedule is based on cosine decay with base learning rate 1e-4, for 100 epochs. In addition, since the target of the outputs of the in-context examples are 0 (see Fig. 1), we optionally add an additional L2 regularizer on the outputs of the in-context examples. See the comparison between the transformers with or without the L2-regularizer in Table 3. In Table 1 of the main paper, the reported numbers correspond to the SL-transformer with the L2 regularizer and the UL-transformer without the L2 regularizer. Across all these settings prefixLM consistently beats causalLM as our theorem predicts.

### E.3    THE IMPACT OF THE SIZE OF THE TRAINING DATA

Here we investigate the performance of prefixLM and causalLM as a function of the number of training samples. In Fig. 6, we plot their after having trained on 10 batches all the way up to 1000 batches (as in Section 5.2). We observe that when the amount of training data is low, ICL falls into the memorization regime, in which models perform perfectly on the training data, but do not generalize well to unseen test sequences. However, prefixLM transitions to the generalization regime earlier than causalLM, which is reflected by the positions of the largest gap between the two. (30 batches in LR, 100 batches in N-LR, and 10 batches in MC).

### E.4    TESTING WITH FEWER IN-CONTEXT EXAMPLES

In causalLM, different positions in the sequence are trained with different numbers of in-context examples (ICEs). This may bring advantage to pretrained causalLM models when tested with fewer number of in-context examples than it was trained on. To compare causalLM and prefixLM in such setting, we use the same models as before that were trained with 40 in-context examples, but test them on fewer (16, 24, 32) in-context examples. Note that 16 is the minimum number of examples to

---

[¶] https://github.com/google-research/google-research/tree/master/incontext

| 16 Test ICEs | LR | N-LR | MC |
|---|---|---|---|
| PrefixLM-SL | 1.01 | 2.1e-2 | 42.8 |
| CausalLM-SL | 1.76 | 2.7e-2 | 43.3 |
| PrefixLM-UL | 0.97 | 1.9e-2 | 42.9 |
| CausalLM-UL | 1.12 | 3.2e-2 | 46.6 |

Table 4: The test query errors with 16 ICEs on linear regression (LR), non-linear regression (NLR), and multiclass classification (MC) tasks.

| 24 Test ICEs | LR | N-LR | MC |
|---|---|---|---|
| PrefixLM-SL | 1.4e-1 | 2.0e-3 | 33.4 |
| CausalLM-SL | 7.0e-1 | 1.0e-2 | 35.9 |
| PrefixLM-UL | 1.0e-1 | 1.7e-3 | 37.1 |
| CausalLM-UL | 1.3e-1 | 1.0e-2 | 41.2 |

Table 5: The test query errors with 24 ICEs on linear regression (LR), non-linear regression (NLR), and multiclass classification (MC) tasks.

solve our 16-dim synthetic regression problems. The errors of prefixLM and causalLM are provided in the following Tables 4, 5, 6, where regression tasks (LR, N-LR) report mean squared errors and the MC task reports the classification error. From the tables we see that prefixLM still consistently outperforms causalLM, even when testing with fewer in-context examples than used during training time.

### E.5 PERMUTATION ON IN-CONTEXT EXAMPLES

We further consider a simple approach for mitigating the problems of causalLM by randomly permuting the in-context examples during training time. This is motivated by the observation that for causalLM, every permutation representations a different view of the context in the example. The results of this experiment (Table 7) show that this style of causalLM training indeed improves over the fixed order training setting compared to the unpermuted ICEs (Table 1). However, prefixLM still outperforms causalLM in general.

### E.6 IN-CONTEXT LEARNING USING PaLM2

Going beyond the publicly available T5 models, we further verify our findings by conducting FLAN-based finetuning experiments using the state-of-the-art PaLM2 family of models (Google et al., 2023). PaLM2 models were pretrained with a mixture of objectives that includes different LM types, which make them a relatively fair starting point to compare causalLM and prefixLM after finetuning. In practice we finetune three sizes of PaLM2 language models: Gecko, Otter and Unicorn[||]. We use the same default recipe for FLAN-PaLM2 finetuning (Google et al., 2023; Chung et al., 2022) and finetune the PaLM2 checkpoints for either causalLM or prefixLM. We then evaluate the ICL capability of the finetuned models on the Massive Multi-task Language Understanding (5-shot MMLU) tasks (Hendrycks et al., 2020).

---

[||]`https://blog.google/technology/ai/google-palm-2-ai-large-language-model/`

| 32 Test ICEs | LR | N-LR | MC |
|---|---|---|---|
| PrefixLM-SL | 2.4e-2 | 4.7e-4 | 32.4 |
| CausalLM-SL | 3.1e-1 | 5.0e-3 | 34.6 |
| PrefixLM-UL | 9.5e-3 | 3.4e-4 | 36.2 |
| CausalLM-UL | 4.0e-2 | 5.7e-3 | 37.3 |

Table 6: The test query errors with 32 ICEs on linear regression (LR), non-linear regression (NLR), and multiclass classification (MC) tasks.

| Permuted ICEs | LR | N-LR | MC |
|---|---|---|---|
| PrefixLM-SL | 9.0e-3 | 1.5e-4 | 24.1 |
| CausalLM-SL | 1.9e-1 | 2.5e-3 | 26.9 |
| PrefixLM-UL | 2.6e-3 | 9.5e-5 | 26.1 |
| CausalLM-UL | 1.1e-2 | 1.8e-3 | 26.2 |

Table 7: The test query errors with randomly permuted ICEs on linear regression (LR), non-linear regression (NLR), and multiclass classification (MC) tasks.

| | Gecko | Otter | Unicorn |
|---|---|---|---|
| PrefixLM | **46.6** | **64.8** | **81.4** |
| CausalLM | 43.3 | 61.0 | 78.0 |

Table 8: The average test query accuracies on 5-shot MMLU tasks with FLAN-finetuned PaLM2-Gecko/Otter/Unicorn prefixLM/causalLM checkpoints. (Google et al., 2023) reported a similar averaged accuracy of 81.2 on Unicorn-PrefixLM.

Table 8 shows that for all three sizes of PaLM2, the MMLU accuracy (average over the 57 tasks) of prefixLM surpasses that of causalLM by more than 3%. This result again empirically verifies that our conjecture generalizes to the practical case, using a state of the art LLM[**].

### E.7 IN-CONTEXT LEARNING WITH MULTIMODAL MODELS

Lastly, we also demonstrate that prefix attention masks benefit ICL in multimodal models across various settings. We conducted experiments using both 4-shot and 8-shot COCO image captioning tasks on the Karpathy split (Karpathy & Fei-Fei, 2015) using the PaLI-X model (Chen et al., 2023), a 55B multimodal pretrained model.

The PaLI-X model employs an encoder-decoder architecture where ViT encoded image tokens and text tokens are fed to the multimodal encoder and decoder to generate outputs. During pretraining, the text prompts were split into two parts. The first part is the input to the multimodal prefix-encoder that self-attends to all the image and text tokens on the encoder side, following the style of prefixLM. The second part is the input to the causal-decoder that self-attends to only the previous text tokens on the decoder side, following the style of causalLM, and cross-attends to encoder tokens.

The prefix-encoder and causal-decoder nature allows us to consider different variants of the attention masks and placements of the in-context texts to showcase the benefits of prefix attention masks. We design two main categories of few-shot experiments with five self-attention mask settings, detailed below. We finetune the PaLI-X pretrained model using each setting's attention mask with 4-shot Episodic WebLI dataset (Chen et al., 2023) for 20k steps.

In the first category, we place the few-shot text tokens on the encoder side and study the effect of manipulating the encoder self-attention masks, leaving the causal-decoder unchanged. Specifically,

---

[**]Besides of PaLM2, we also find that any checkpoint that is pretrained with a mixture of prefixLM and causalLM tends to do better with prefixLM for in-context learning. However, we do not claim that prefixLM would necessarily outperform causalLM when using solely causalLM pretrained checkpoints.

| | 4-shot | 8-shot |
|---|---|---|
| Prefix encoder | **106.7** | **107.5** |
| Block-causal encoder | 104.8 | 106.0 |
| Causal encoder | 102.3 | 104.9 |
| Prefix decoder | **103.9** | **104.2** |
| Causal decoder | 102.4 | 92.9 |

Table 9: Cider scores of COCO captioning using various attention masks. The Prefix variant outperforms the Causal ones. Note that the official PaLI-X (Chen et al., 2023) reported a 4-shot Cider of 107.6, which was also based on the prefix encoder mask, but was finetuned with additional image captioning data from the Conceptual Captions 3M dataset (Sharma et al., 2018).

considering a 2-shot ICL case for simplicity, we adapt the prefix encoder attention mask $A_{prefix}^{enc}$ in (19) into two causal variants, block-causal and causal encoder attention masks as $A_{b-causal}^{enc}$ in (20) and $A_{causal}^{enc}$ in (21). In this case, the block-causal version is more inline with exposing the encoder to the examples one at a time, while the causal one strictly follows auto-regressive attention.

$$
A_{prefix}^{enc} = 
\begin{array}{ccccc}
I_1 & T_1 & I_2 & T_2 & I_t
\end{array}
\left(
\begin{array}{ccccc}
\mathbb{1} & \mathbb{1} & \mathbb{1} & \mathbb{1} & \mathbb{1} \\
\mathbb{1} & \mathbb{1} & \mathbb{1} & \mathbb{1} & \mathbb{1} \\
\mathbb{1} & \mathbb{1} & \mathbb{1} & \mathbb{1} & \mathbb{1} \\
\mathbb{1} & \mathbb{1} & \mathbb{1} & \mathbb{1} & \mathbb{1} \\
\mathbb{1} & \mathbb{1} & \mathbb{1} & \mathbb{1} & \mathbb{1}
\end{array}
\right)
\begin{array}{c}
I_1 \\ T_1 \\ I_2 \\ T_2 \\ I_t
\end{array}
\tag{19}
$$

$$
A_{b-causal}^{enc} = 
\begin{array}{ccccc}
I_1 & T_1 & I_2 & T_2 & I_t
\end{array}
\left(
\begin{array}{ccccc}
\mathbb{1} & \mathbb{1} & \mathbb{1} & \mathbb{1} & \mathbb{1} \\
\mathbb{1} & \mathbb{1} & \mathbb{1} & \mathbb{1} & \mathbb{1} \\
 & & \mathbb{1} & \mathbb{1} & \mathbb{1} \\
 & & \mathbb{1} & \mathbb{1} & \mathbb{1} \\
 & & & & \mathbb{1}
\end{array}
\right)
\begin{array}{c}
I_1 \\ T_1 \\ I_2 \\ T_2 \\ I_t
\end{array}
\tag{20}
$$

$$
A_{causal}^{enc} = 
\begin{array}{ccccc}
I_1 & T_1 & I_2 & T_2 & I_t
\end{array}
\left(
\begin{array}{ccccc}
\mathbb{1} & \mathbb{1} & \mathbb{1} & \mathbb{1} & \mathbb{1} \\
 & \diagdown & \mathbb{1} & \mathbb{1} & \mathbb{1} \\
 & & \mathbb{1} & \mathbb{1} & \mathbb{1} \\
 & & & \diagdown & \mathbb{1} \\
 & & & & \mathbb{1}
\end{array}
\right)
\begin{array}{c}
I_1 \\ T_1 \\ I_2 \\ T_2 \\ I_t
\end{array}
\tag{21}
$$

$I_1, I_2, I_t$ denotes the image tokens for the two shots and the target and $T_1, T_2$ denotes the text tokens for the two shots. $\mathbb{1}$ denotes a matrix of all 1s and "$\diagdown$" denote an upper triangular matrix with 1s. A 1 at row $i$ and column $j$ indicates that token $j$ is allowed to attend to token $i$. We report the results on few-shot COCO captioning in the top half of Table 9. We observe consistent improvement over both 4- and 8-shot when changing the encoder attention mask from causal mask, to block causal mask, and then to prefix mask.

$$
A_{causal}^{dec} = 
\begin{array}{ccc}
T_1 & T_2 & T_t
\end{array}
\left(
\begin{array}{ccc}
\diagdown & \mathbb{1} & \mathbb{1} \\
 & \diagdown & \mathbb{1} \\
 & & \diagdown
\end{array}
\right)
\begin{array}{c}
T_1 \\ T_2 \\ T_t
\end{array}
\tag{22}
$$

$$
A_{prefix}^{dec} = 
\begin{array}{ccc}
T_1 & T_2 & T_t
\end{array}
\left(
\begin{array}{ccc}
\mathbb{1} & \mathbb{1} & \mathbb{1} \\
\mathbb{1} & \mathbb{1} & \mathbb{1} \\
 & & \diagdown
\end{array}
\right)
\begin{array}{c}
T_1 \\ T_2 \\ T_t
\end{array}
\tag{23}
$$

Similarly, in the second category, we place the few-shot text on the decoder side and study the effect of manipulating the decoder attention masks, leaving the prefix encoder unchanged. We adapt the causal decoder attention mask $A_{causal}^{dec}$ in (22) to prefix attention mask $A_{prefix}^{dec}$ in (23). Note that all the image tokens from the prefix-encoder side are visible to all text tokens (on the decoder) via cross attention. However, the image tokens cannot attend to the text because of the encoder-decoder architecture. The second half of Table 9 reports the results of using prefix and causal decoder attention. Even though the decoder is pretrained in the causal manner, with additional finetuning using prefix masks, the new prefix decoder achieves a Cider score of 103.9 in 4-shot ICL, outperforming the finetuned causal decoder by 1.5. Furthermore, the prefix decoder also appears to be more robust when extrapolating to 8-shot evaluation (Cider 104.2), compared to the causal decoder (Cider 92.9).

In summary, the LLM experiments in Section 5.3 as well as the multimodal experiments in this section show that our conjectures hold up in practice with various types of large-scale models and a wide range of settings.