# OpenReview forum: "CausalLM is not optimal for in-context learning"
_ICLR.cc/2024/Conference — ICLR 2024 poster_

### Official Review · Reviewer_mzWe · 2023-10-28

**Soundness:** 3 good
**Presentation:** 3 good
**Contribution:** 4 excellent
**Rating:** 8
**Confidence:** 4

**Summary:**

This paper analyzes the convergence behavior of in-context learning under a certain parameter construction and shows that while both LM types converge to their stationary points at a linear rate, prefixLM converges to the optimal solution of linear regression while causalLM may include a bias that may not diminish even if the number of samples grows infinitely. They also show with experiments on both synthetic and real tasks that prefixLM seems to outperform causalLM given the same amount of parameters.

**Strengths:**

* **Originality.** The paper reveals the surprising finding that CausalLM may underperform PrefixLM in the task of in-context learning.

* **Significance.** The paper is of significance in understanding the in-context learning mechanism.

* **Clarity.** The paper is well-written and easy to read.

* **Quality.** The experiments are thorough and the theory is rigorous.

**Weaknesses:**

* To the reviewer's understanding, this paper hasn't excluded the possibility that there exists a parameter configuration that can allow CausalLM to reach optimal solution in the linear setting.

**Questions:**

* **Q1.** (See weakness) If the reviewer's understanding is correct, then a question is whether proving such a statement is possible.

* **Q2.** Is it possible to convert a pretrained CausalLM into a PrefixLM during in-context learning through fine-tuning and improve the performance of ICL?

---

> ### Author Response · Authors · 2023-11-22
>
> Thank you for your questions. Please see our answers to the questions.
>
> 1. In a one-layer LSA, it can be proved (using the method in [1]) that the parameter configuration that we used in the paper is optimal for both prefixLM and causalLM in linear regression ICL. Although the proof does not necessarily generalize to the multi-layer case, we strongly suspect that, due to its architectural constraints, there is no parameter configuration that would allow causalLM to reach optimal in the linear setting given a finite number of examples.
> Specifically, the causalLM architecture causes the model to process the data in an online learning manner. An online learning may asymptotically converge to the optimum given infinitely many examples, when the step size of the learning algorithm is decreasing (which may be possible with a different LSA parameter configuration). However, even if an online learning system asymptotically converges to the optimal solution, it can be proved that at least O(1/\eps) examples are needed in order to reach the \eps-ball of the solution, which is still suboptimal compared to a batch optimizer when the number of examples is limited.
>
> [1] Ruiqi Zhang, et al. Trained transformers learn linear models in-context. arXiv preprint arXiv:2306.09927, 2023.
>
> 2. In Appendix E.5, we study replacing the attention masks of a pretrained causal decoder in the PaLI-X model by a prefix decoder during finetuning. In our experiment, we found that the prefix-decoder improved the test accuracy over the original causal-decoder in both 4-shot and 8-shot cases. However, since such a swap introduces an architecture gap between pretraining and finetuning, we cannot guarantee that is always beneficial. On the other hand, we find that as far as the pretraining mixture contains both prefixLM and causalLM objectives, the prefixLM always outperforms causalLM on ICL.

---

### Official Review · Reviewer_m1Xp · 2023-10-31

**Soundness:** 3 good
**Presentation:** 3 good
**Contribution:** 3 good
**Rating:** 6
**Confidence:** 3

**Summary:**

The paper argues that in context learning with causalLM is inferior to in context learning with prefixLM. They extend the theory of Von Oswald et al. to include multi-layer Linear Self-Attention and multi-step gradient descent in the setup considered in Von Onswald et al., showing convergence result in the extended setup for both causalLM and prefixLM in context learning. Using their developed theory and experimental validation, they demonstrate that prefixLM is superior to causalLM.

**Strengths:**

The paper follows an emerging line of papers showing the equivalence of gradient descent to in context learning in a very specific setup where the self-attention is linear, the objective is linear regression and the parameter matrices are hand-constructed.

Abstracting away the limited setup, the paper does a good job extending the theory of Von Onswal et al. The theoretical argument is clear in my opinion and the evidences supporting the main thesis of the work are convincing enough. In particular, the paper does a good job arguing in details why prefixLM is better to causalLM both theoretically and empirically on more realistic setups.

**Weaknesses:**

My understanding is that this work aims at demonstrating that prefixLM is superior to causalLM for in context learning. While I believe they do a good job at it, I am under the impression that most projects already used prefixLM when possible. For example, InstructBLIP and Llama2 use prefixLM as far as I can understand. If this is the case that most influential language model or VLM already use prefixLM then I am unclear about the intended impact of this work.

The models used in this work are internal models that cannot be accessed by anyone outside Google. I believe that this is a big problem for open science and reproducible research. Unless the conclusions made in this work cannot be attained with available open models, which I highly doubt, I urge the authors to consider using open alternative to disseminate their results.



Minor:
* The considered theoretical setup seems contrived to me.  Since I don't expect the author to come up with a new theoretical setup, this point is not reflected in my evaluation of this paper. However, I believe that this is a weakness of this line of works and I would appreciate to be convinced why any conclusion made on this setup (LSA, linear regression and hand-crafted parameters) should necessary translate to more realistic setups.
* Figure 1 is blurry on printed paper.

**Questions:**

* Does prior works generally use causalLM instead of prefixLM?
* Are the empirical results presented in this work reproducible with open models?

---

> ### Author Response · Authors · 2023-11-22
>
> Thank you for your questions. Please see our answers to your questions.
>
> 1. Although more and more recent models (especially the multimodal ones) are starting to apply the prefixLM (or encoder-decoder) architecture, there are still prominent models which do not, e.g. Flamingo, GPT-family, etc. Therefore, we believe that the comparison between causalLM and prefixLM remains important. Furthermore, while prefixLM is being adopted by many models recently, that line of work has not provided rigorous proofs that explain why prefixLM should be better than causalLM. To our best knowledge, our paper is the first to rigorously argue that from a theoretical perspective (in particular for multi-layer transformer ICL).
>
>
> 2. Following the reviewer’s concerns about the reproducibility of our results, we conduct a new set of large-scale experiments using the open source t5 models, which we describe in detail below. At the same time, please note that our first two experiments in Sec 5.1, 5.2 compare causalLM and prefixLM in a small experimental setup with synthetic data, and try to directly verify our theorems. The work done in these sections is readily reproducible for people with access to the jax/flax library. We will release the code to reproduce these main synthetic results of our paper after the anonymous period.
>
> For the large scale experiments with LM or VLM, a pretraining procedure is needed to get reasonable results. To set up the experiments for comparing prefixLM and causalLM in a fair manner, two options are available: (A) two pretrained models that only differ in its attention masks (causalLM vs prefixLM) or (B) one pretrained model which was exposed to both causalLM and prefixLM during pretraining.
>
> At the time of submission, there were no type-A models available, and the PaLM2 model was the only type-B model that we had access to. Unfortunately, it was not released and cannot be used for reproducibility.
>
> In order to make sure that our main empirical results hold and are reproducible, we conducted another experiment based on the t5 models and c4 corpus (both publicly available). Note that the existing public t5 checkpoints are all based on EncDec models, which are similar to prefixLM. Thus, it would be unfair and unnatural to compare with causalLM by simply replacing the bidirectional attention of the encoder to the causal attention during the finetuning stage.
>
> To make a more reasonable comparison, we reran the pretraining stages of t5 on the c4 corpus but with DecoderOnly models. Moreover, for each size (from base, large and xl) of the models, we pretrained two checkpoints, one with prefixLM and the other with causalLM, each for 1M steps using the same t5 pretraining recipe. In this way, we construct a type-A fair comparison between prefixLM and causalLM.
>
> After pretraining, we use the FLAN recipe to finetune each checkpoint (for 20k steps) with its pretrained attention mask and compare them on two benchmarks: MMLU [1] and BBH [2], under the few-shot ICL setting in [3]. Their averaged accuracies are reported in the following tables:
>
> |MMLU Avg. Acc (higher is better) | Base | Large | XL |
> |--- |---|---|---|
> |PrefixLM | 26.1 | 31.9 | 39.0 |
> |CausalLM | 26.6 | 26.1 | 30.2 |
>
> |BBH Avg. Acc (higher is better) | Base | Large | XL |
> |--- |---|---|---|
> |PrefixLM | 26.6 | 30.2 | 34.0 |
> |CausalLM | 23.8 | 28.3 | 29.4 |
>
> As can be seen, except for the base-size MMLU case, where causalLM marginally wins over prefixLM, in all other five cases prefixLM wins by a significant margin. Furthermore, as the model size gets larger, the advantage of prefixLM tends to grow larger.
> We want to specifically thank the reviewer for pushing us in this direction, as we consider that these experiments strengthen the claims of our paper. We plan to add this set of new results into our main body of the paper, and will move the PaLM2 results into the appendix. Furthermore, we will release the corresponding checkpoints to make sure the above results are reproducible.
>
> [1] Dan Hendrycks, et al. Measuring massive multitask language understanding. ICLR, 2020.
>
> [2] Mirac Suzgun, et al. Challenging BIG-Bench tasks and whether chain-of-thought can solve them. arXiv preprint arXiv:2210.09261, 2022.
>
> [3] Hyung Won Chung, et al. Scaling instruction-finetuned language models. arXiv preprint arXiv:2210.11416, 2022.
>
>
> Minor: We agree that the theoretical setup has its limitations, as the exact mathematical formalism is hard to generalize beyond LSA. Nevertheless, we use this simplified setup as a probe, to reveal an important consequence of the architecture differences, which could generalize to more general transformers. Specifically, the attention mask of prefixLM allows it to do batch learning, while the one of causalLM restricts it to online learning, which is suboptimal for few-shot learning. Our simplified setup allows us to conduct more rigorous analysis through the stationary points, so that such comparison is more obvious.

---

> > ### Comment · Reviewer_m1Xp · 2023-12-02
> > **Thank you for running the experiments on open models**
> >
> > I appreciate the efforts put by the authors to make their finding reproducible by the community.

---

### Official Review · Reviewer_RFAv · 2023-11-01

**Soundness:** 3 good
**Presentation:** 3 good
**Contribution:** 2 fair
**Rating:** 6
**Confidence:** 3

**Summary:**

The paper examines transformer-based in-context learning using pre-fixed language models (pre-fixedLM) and causal language models (causalLM). While pre-fixedLMs are empirically observed to outperform causalLMs, the theoretical reasons remain elusive. Through a convergence analysis, the study reveals that pre-fixedLMs optimally converge to linear regression solutions, whereas causalLMs follow online gradient descent dynamics, leading to suboptimal results. Empirical tests validate these theoretical findings.

**Strengths:**

1. Theoretical understanding of different solutions found by prefix and casual LMs are provided under the linear regression setting, which seems novel.
2. The paper also provides an empirical study to compare the solution found by prefix and causal LMs in different tasks, which verifies the theoretical intuitions.

**Weaknesses:**

1. The experimental setting is limited to a given number of in-context examples which seems to naturally favor prefix LMs. Casual LMs would train the model with different numbers of in-context samples simultaneously while prefix LM using all possible in-context and query partitions with the same in-context length. Testing with fewer in-context examples could be beneficial to provide more comprehensive results.
2. Another unfairness in the experimental setting is that the nature of casual LMs would basically reduce the number of examples with the same in-context samples compared to prefix LM (i.e. for one input sequence, prefix LMs would enumerate all possible in-context and query partitions but casual LM would only have one for a specific context length). See Q1.

**Questions:**

Q1. A simple augmentation could be used to manually feed those other possibilities of partitions to casual LM, so prefix LM and casual LM would be trained with the same number of in-context and query possibilities for a given context length. Under this setting, would prefix LM still perform better than casual LM?

---

> ### Author Response · Authors · 2023-11-22
>
> Thank you for your questions and the constructive suggestions on the two ablation tests. Please check our following experimental results to see whether they address your concerns.
> 1. We follow your suggestion by using the models that were trained with 40 in-context examples, but testing them on fewer (16, 24, 32) in-context examples. Note that 16 is the minimum number of examples to solve our 16-dim synthetic regression problems. The errors of prefixLM and causalLM are provided in the following table, where regression tasks (LR, N-LR) report mean squared errors; and the MC task reports the classification error. We see that in general prefixLM still consistently outperforms causalLM even when testing with a smaller number of in-context examples than the training time, a setting that favors causalLM.
>
> |16 test ICEs | LR (MSE) | N-LR (MSE) | MC (classification error) |
> |---|---|---|---|
> |PrefixLM-SL |1.01 | 2.1e-2 | 42.8 |
> |CausalLM-SL | 1.76 | 2.7e-2 | 43.3 |
> |PrefixLM-UL | 0.97 | 1.9e-2 | 42.9 |
> |CausalLM-UL | 1.12 | 3.2e-2 | 46.6 |
>
> |24 test ICEs | LR (MSE) | N-LR (MSE) | MC (classification error) |
> |---|---|---|---|
> |PrefixLM-SL |1.4e-1 | 2.0e-3 | 33.4 |
> |CausalLM-SL | 7.0e-1 | 1.0e-2 | 35.9 |
> |PrefixLM-UL | 1.0e-1 | 1.7e-3 | 37.1 |
> |CausalLM-UL | 1.3e-1 | 1.0e-2 | 41.2 |
>
> |32 test ICEs | LR (MSE) | N-LR (MSE) | MC (classification error) |
> |---|---|---|---|
> |PrefixLM-SL |2.4e-2 | 4.7e-4 | 32.4 |
> |CausalLM-SL | 3.1e-1 | 5.0e-3 | 34.6 |
> |PrefixLM-UL | 9.5e-3 | 3.4e-4 | 36.2 |
> |CausalLM-UL | 4.0e-2 | 5.7e-3 | 37.3 |
>
> 2. To address your concern, we conduct a new set of experiments in which each sequence of data permutes the in-context examples in random order during the training time.
> The results of this experiment show that this style of causalLM training indeed improves over the fixed order training setting. However, prefixLM still outperforms causalLM in general. We will add these informative experiments to the appendix and discuss how for simple tasks, permutation of in-context examples might mitigate some of the problems with causalLM.
>
> | Permuted ICEs | LR (MSE) | N-LR (MSE) | MC (classification error) |
> |---|---|---|---|
> |PrefixLM-SL |9.0e-3 | 1.5e-4 | 24.1 |
> |CausalLM-SL | 1.9e-1 | 2.5e-3 | 26.9 |
> |PrefixLM-UL | 2.6e-3 | 9.5e-5 | 26.1 |
> |CausalLM-UL | 1.1e-2 | 1.8e-3 | 26.2 |

---

### Official Review · Reviewer_9q6M · 2023-11-04

**Soundness:** 3 good
**Presentation:** 3 good
**Contribution:** 2 fair
**Rating:** 6
**Confidence:** 4

**Summary:**

In this work, authors relate the in-context learning abilities of causalLM architectures with that of an online gradient descent solver, versus the abilities of a prefixLM architecture that show convergence to the optimal solution at a linear rate. I found the experiments well conducted and the paper's results well presented overall. I'm unsure of the problem's relevance, but find the paper interesting regardless.

**Strengths:**

Primarily a study of ICL's limitations in standard transformer architectures, which use causal self-attention masking and hence causal language modeling objectives. prefixLMs' superiority is also described and the empirical results are interesting.

**Weaknesses:**

While I find the results and the paper itself interesting, as I mentioned in my summary of the paper, I'm unsure of the relevance of the problem. I think it would help improve the paper's impact if the motivations were clarified better: specifically, what models actually use a PrefixLM architecture in current literature and demonstrating or citing papers which show these models have a different qualitative behavior that a standard causalLM architecture.

**Questions:**

See weaknesses

---

> ### Author Response · Authors · 2023-11-22
>
> Thank you for your question about the relevance of the problem.
>
> The interest of the NLP/ML community in the comparison between prefixLM and causalLM can be traced back as early as [1], where they showed prefixLM outperforms causalLM in varieties of NL tasks. Later, UL-2 [2] proposed to mix prefixLM and span corruption objectives, and found it to be more efficient than the causalLM objective alone. It was also shown in [3], that U-PaLM (a UL2-finetuning PaLM) outperforms PaLM (causalLM only) in various ICL tasks. Indeed, for the reasons above, some of the latest models have included prefixLM objectives in the pretraining mix (for example PaLM-2 [4]). On the other hand, prominent models such as Flamingo as well as the ones in the GPT-family are still based on the causalLM structure, so the comparison between prefixLM and causalLM remains important and relevant. Furthermore, all previous studies were done in an empirical manner, whereas we set out to explain their differences from a theoretical perspective and back the theory with empirical evidence. While we are not the first to follow this path, our work is the first to provide a theoretical justification for the advantage of prefixLM over causalLM in a multi-layer transformer ICL setting by analyzing their theoretical convergence properties.
> We will edit the paper to clarify the above points.
>
> [1] Colin Raffel, et al. Exploring the limits of transfer learning with a unified text-to-text transformer. The Journal of Machine Learning Research, 21(1):5485–5551, 2020.
>
> [2] Yi Tay, et al. Unifying language learning paradigms. arXiv preprint arXiv:2205.05131, 2022.
>
> [3] Hyung Won Chung, et al. Scaling instruction-finetuned language models. arXiv preprint arXiv:2210.11416, 2022.
>
> [4] Google, et al. Palm 2 technical report, 2023.

---

### Meta-Review · Area_Chair_a8j9 · 2023-12-06

**Metareview:**

This paper contributes by demonstrating the superior performance of transformer-based in-context learning with a prefix language model (prefixLM) compared to causal language models (causalLM). The empirical evidence and theoretical analysis show that prefixLM converges optimally, while causalLM follows suboptimal convergence dynamics. The consistent underperformance of causalLM is confirmed through empirical experiments across various tasks. The empirical results are potentially significant contributions to the community. I recommend accepting this paper. As noted by the reviewers, it would be crucial for the authors to release all source codes and experimental details to facilitate the reproduction of the results by others.

**Justification For Why Not Higher Score:**

This paper makes a solid contribution, but the idea is not revolutionary in the area.

**Justification For Why Not Lower Score:**

This paper makes a solid contribution to the area. The technical quality is good and the presentation is clear.

---

### Decision · Program_Chairs · 2024-01-16

Accept (poster)